# The Efficiency of Different Priming Agents for Improving Germination and Early Seedling Growth of Local Tunisian Barley under Salinity Stress

**DOI:** 10.3390/plants10112264

**Published:** 2021-10-22

**Authors:** Rim Ben Youssef, Nahida Jelali, Nadia Boukari, Alfonso Albacete, Cristina Martinez, Francisco Perez Alfocea, Chedly Abdelly

**Affiliations:** 1Laboratory of Extremophile Plants, Biotechnology Center of Borj-Cedria, P.O. Box 901, Hammam-Lif 2050, Tunisia; nahidajelali@gmail.com (N.J.); nedia.boukari@yahoo.fr (N.B.); abdelly.chedly@gmail.com (C.A.); 2Faculty of Sciences of Tunis, University of Tunis El Manar, Tunis 1060, Tunisia; 3Centro de Edafología y Biología Aplicada del Segura, Spanish National Research Council (CEBAS-CSIC), Departameno Nutricion Vegetal, 30100 Murcia, Spain; alfonsoa.albacete@carm.es (A.A.); cmandujar@cebas.csic.es (C.M.); alfocea@cebas.csic.es (F.P.A.)

**Keywords:** pretreatment, salt stress, *Hordeum vulgare* (L. Manel), *Hordeum maritimum*, CaCl_2_, KCl, KNO_3_

## Abstract

The current work aimed to investigate the effect of seed priming with different agents (CaCl_2_, KCl, and KNO_3_) on germination and seedling establishment in seeds of the barley species of both *Hordeum vulgare* (L. Manel) and *Hordeum maritimum* germinated with three salt concentrations (0, 100, and 200 mM NaCl). The results showed that under unprimed conditions, salt stress significantly reduced the final germination rate, the mean daily germination, and the seedling length and dry weight. It led to a decrease in the essential nutrient content (iron, calcium, magnesium, and potassium) against an increase in sodium level in both of the barley species. Moreover, this environmental constraint provoked a membrane injury caused by a considerable increase in electrolyte leakage and the malondialdehyde content (MDA). Data analysis proved that seed priming with CaCl_2_, KCl, and KNO_3_ was an effective method for alleviating barley seed germination caused by salt stress to varying degrees. Different priming treatments clearly stimulated germination parameters and the essential nutrient concentration, in addition to increasing the seedling growth rate. The application of seed priming reduced the accumulation of sodium ions and mitigated the oxidative stress of seeds caused by salt. This mitigation was traduced by the maintenance of low levels of MDA and electrolyte leakage. We conclude that the priming agents can be classed into three ranges based on their efficacy on the different parameters analyzed; CaCl_2_ was placed in the first range, followed closely by KNO_3_, while the least effective was KCl, which placed in the third range.

## 1. Introduction

Germination, as the most crucial process in the crop life cycle, is regulated by both internal and external factors. Internal factors include proteins, plant hormones, and seed age and size. External factors incorporate abiotic and biotic constraints such as drought, temperature, salinity, nutritional deficiency, metal, and pathogen disease. Germination is extremely sensitive to these environmental stressors [1]. Salinity is among the most harmful environmental stresses that restrict agricultural productivity [2]. It greatly limits crop yield in semi-arid and arid regions, and affects approximately 397 million hectares of fertile soil in the world [3]. In fact, under such an environmental stressor, seedling growth and the productivity of several crops are affected due to water limitations and ion imbalance [4,5]. A negative correlation has also been found with seed germination, as the toxic effect of sodium and chloride seriously affect embryo viability [6]. In addition, Patade et al. [7] and Ansari and Sharif-Zadeh [8] reported that salinity not only decreases the germination percentage and prolongs seed dormancy by delaying the start of germination, but also leads to the death of seeds before germination.

It is well known that in the Mediterranean region, the majority of plant species are very sensitive to salt stress. For this reason, adaptations and improving seed tolerance to salinity during germination represent a particular agronomic goal and are important for the economics of many countries worldwide. Several strategies for improving the growth and development of crops have been investigated for many years. These approaches are based on old, costly, and slow-moving techniques such as the selection of species (screening). As a solution to overcome these adverse conditions in agricultural land, seed priming is one of the most frequently used techniques employed by farmers. It is based on the pre-exposure of seeds or young seedlings to chemical agents or to abiotic stressors (such as salinity, drought, and nutrient deficiency), making them more resistant to subsequent stresses and enhancing their ability to detect secondary signals quickly [9]. Recently, Boukari et al. demonstrated that seed priming ameliorates the growth of alfalfa plants subjected to salinity, iron deficiency, and their combined effect [10]. It has also been reported that seed pretreatment enhances the tolerance of many other crops to salt stress by stimulating seed germination [11,12], such as lettuce [13], tomato [14], and maize [15,16].

Many efficient compounds are used to broaden the application of this technique. Among them, potassium and calcium ions are used for osmo priming through treatment with KNO_3_, KCl, and CaCl_2_. These compounds are relatively inexpensive and easy to obtain [17], and several studies have reported that they are effective at promoting seed germination under salt stress [12,13,14,15,16,17,18]. Indeed, it has been shown that CaCl_2_ plays a crucial role in regulating plant hormone metabolism [19]. Moreover, recent reports suggest that the ameliorative effect of external calcium on plants facing salinity may be associated with the maintenance of an optimal sodium/potassium ratio in the cytosol [20,21]. Similar interest has been focused on the effect of potassium nitrate (KNO_3_) seed priming in several plant species. As proven by Joshi et al., treating seeds with KNO_3_ ameliorates tolerance in cucumber to salt stress by activating the antioxidant system and accumulating proline [22]. Generally, during germination under salt stress, seeds treated with KNO_3_ exhibit an increase in proteins, free amino acids, and soluble sugars [23]. Additionally, Kaya et al. revealed that supplying KNO_3_ had a key role in regulating osmotic pressure and activating enzymes [24]. Furthermore, several studies demonstrated the alleviating role of KCl during salinity by reducing the sodium levels. Indeed, KCl priming improved safflower seed germination under salinity [25]. Furthermore, the findings of Ruan et al. demonstrated that the application of KCl seed priming to rice had improved results for the germination index [26]. In this regard, Taiz and Zeiger related the efficiency of KCl priming to the osmotic power of potassium ions to enhance cell water saturation [27].

*Hordeum maritimum* is considered an important wild cereal and as an annual facultative halophyte in Tunisian flora, widely distributed mainly in saline habitats such as Soliman and Kalbia Sebkhas, characterized by an electrical conductivity of saturated soil paste extract (ECe) around 19.0 dSm^−1^ and a soluble sodium content of about 92.6 µmolg^−1^ soil, corresponding to slightly more than 200 mM [28]. *Hordeum maritimum* contributes significantly to the annual biomass production in Tunisia because it is able to tolerate high salt concentrations without showing symptoms of toxicity (it can survive up to 300 mM NaCl in saline zones) [29,30]. It deals with these adverse conditions by developing appropriate physiological responses such as free proline, glycine betaine, and free amino acids through a process known as osmotic adjustment in response to the decreased external water potential [31]. *Hordeum vulgare* (L. Manel) is a cereal grain and a glycophyte that ranks fifth among all crops for dry matter production worldwide. It is considered an important food source, especially in Tunisian ecosystems.

*Hordeum maritimum*, a wild seaside barley, is a halophyte plant, while *H.vulgare* is a glycophyte, and their salt tolerance mechanisms are completely different. Indeed, wild barley has a more pronounced salt-tolerant behavior than cultivated barley [32]. For this reason, H. *maritimum* is an appropriate model to study the mechanisms of salinity tolerance in cereals [33]. In fact, when soil salinity is a temporary condition, glycophytes, a group of plants to which the majority of crops belong, have to cope with osmotic stress and ion toxicity using specific short-term strategies that allow for osmotically balancing the cell, control ion, and water homeostasis, and recover stress induced damage based on the production of a low osmolyte content, as well as the possibility of a reduction in the salt concentration in the root medium [34]. These adaptation mechanisms are of central importance in glycophytes allowing for the ability to grow under stressful conditions [35]. However, the defense mechanisms that halophyte plants have evolved to adapt to salt stress are mainly partitioning sodium and chloride in the vacuole, as well as reserving the synthesis and accumulation of organic solutes for a high salinity (200 and 300 mM NaCl)in the cytosol [35]. In particular, in many halophytes, proline and glycine betaine are highly accumulated and contribute to the balance of osmotic pressure in the cell as a whole [36]. These compatible compounds protect macromolecular structures and mitigate oxidative damage adjustment, in addition to their fundamental role of osmotically balancing the cytosol [37]. Even more resistant than other cereal crops to environmental constraints, when grown in saline soils, cultivated barley, has a lower degree of tolerance compared with wild species. The higher degree of salt tolerance of *H. maritimum* compared with *H. vulgare* is related to its higher membrane integrity. In fact, the levels of membrane lipid peroxidation and electrolyte leakage increased considerably with a lower salt concentration in *H. vulgare* because of oxidative damage, whereas those of *H. maritimum* changed remarkably only under high salt concentrations [38].

Hence, *H. vulgare* and *H. maritimum* are both species of considerable interest, and it would be valuable to perform more detailed studies in order to improve their tolerance to various environmental constraints. Although there are many reports considering the beneficial effects of seed priming agents in the germination and growth of many crops during salt stress, no previous work has mainly focused on their impact in the germination stage of both of these local Tunisian barley species. Furthermore, investigations comparing the relative effects of different priming agents on barley germination under different salt treatments have rarely been reported. For this reason, this study intended to clarify the role of priming seeds with KNO_3_, KCl, or CaCl_2_ for improving the tolerance of the barley species of both *H. vulgare* (L. Manel) (cultivated) and *H. maritimum* (wild) when subjected to salinity.

## 2. Results

### 2.1. Effect of Seed Priming on Germination Parameters

#### 2.1.1. Final Germination Rate (FG %)

The results of Figure 1 show that priming with KCl, KNO_3_, or CaCl_2_ alleviated the impact of salt stress caused by either 100 or 200 Mm NaCl on the FG% of the barley seeds. Among the priming agents, CaCl_2_ seemed to be the most efficient protector, causing an increase of 26% and 61% in FG% in *H. maritimum* and *H. vulgare* (L. Manel), respectively, particularly when the seeds were exposed to 200 mM NaCl compared with their unprimed equivalents, while in the absence of priming treatments, seed germination was clearly affected by the salinity. The application of 200 mM NaCl caused a 34% and 47% reduction in FG% for *H. maritimum* and *H. vulgare* (L. Manel), respectively, compared with the control.

#### 2.1.2. Mean Daily Germination (MDG)

The results shown in Figure 2 illustrate that without seed priming, salt stress significantly decreased MDG in both barley species. This decrease was more obvious when *H. vulgare* (L. Manel) seeds were exposed to 200 mM NaCl. Interestingly, priming with KCl, KNO_3_, or CaCl_2_ promoted seed germination by increasing the values of MDG in *H. maritimum* and *H. vulgare* (L. Manel) seeds subjected either to 100 or 200 mM NaCl. The beneficial effect of several agents was similar and clearer in *H. vulgare* (L. Manel).

### 2.2. Effect of Seed Priming on Seedling Growth

#### 2.2.1. Seedling Length

The data of Figure 3 show that both NaCl concentrations negatively affected seedling length in both barley species. For unprimed seeds of *H. vulgare* (L. Manel), the coleoptiles and radicles length decreased by 67% and 65%, respectively, while, those of *H. maritimum* diminished by 37% and 27%, respectively, when they were exposed to 200 mM NaCl, compared with their untreated controls. However, among all of the priming agents applied, CaCl_2_ mostly enhanced the growth of the radicle and the coleoptile length in both species, with or without salt treatment. In fact, when the seeds were exposed to 200 mM NaCl, the stimulation of radicle length reached 92% and 37%, respectively; however, the coleoptile length was induced by 140% and 32%, respectively in *H. vulgare* (L. Manel) and *H. maritimum,* respectively, compared with their unprimed stressed equivalents.

#### 2.2.2. Dry Biomass

Salinity had a significant inhibitory effect on the dry biomass of the unprimed seedlings of both barley species (Figure 4). This effect was more pronounced in *H. vulgare* (L. Manel) than in *H. maritimum*. Indeed, when the seeds were subjected to 200 mM NaCl, the coleoptile dry weight reached a decrease of 61% and 35%, respectively, also, radicles dry weight was reduced by 61.5% and 38%, respectively in *H. vulgare* (L. Manel) and *H. maritimum* compared to the control. Nevertheless, seed priming with CaCl_2_, KCl and KNO_3_ considerably alleviated the adverse effect of salinity by promoting seedling dry biomass of both species. It is noteworthy that among these priming agents, CaCl_2_ seemed to be the most efficient protector under either 100 mM or 200 mM NaCl (Figure 4). It stimulated the radicle dry weight of the seedlings from the seeds subjected to 200 mM by 94% and 43% in *H. vulgare* (L. Manel) and *H. maritimum,* respectively, compared with the seedlings from the stressed unprimed seeds. CaCl_2_ clearly increased the coleoptile dry weight in *H. vulgare* (L. Manel) germinated with 100 mM NaCl by 114%, while the rate of increase reached only 30% in *H. maritimum* exposed to 200 mM NaCl, compared with their unprimed equivalents.

#### 2.2.3. Mineral Nutrition

##### Iron

The data presented in Table 1 demonstrate that for the seedlings from unprimed seeds, salt treatments significantly decreased the iron content in both barley species. Under the effect of 200 mM NaCl, the iron concentration was reduced by 71.7% and 38.5% in coleoptile, while in radicle the reduction rate was about 56% and 34.9% in *H. vulgare* (L. Manel) and *H. maritimum*, respectively, compared with the untreated control. Interestingly, all of the priming agents positively affected the iron content of the seedlings from the stressed seeds of both species compared with the unprimed stressed ones. However, this beneficial effect was clearly marked in the *H. vulgare* (L. Manel) species. When the seeds of *H. vulgare* (L. Manel) were exposed to 200 mM NaCl, the best stimulation of the Fe concentration was recorded in coleoptile, which reached 90% in the seedlings from the seeds pretreated with CaCl_2_ and 89% in the seedlings from seeds pretreated with KNO_3_, while it was about 79% and 66% in the radicle of seedlings from pretreated seeds with CaCl_2_ and KNO_3_, respectively, compared with their unprimed equivalents.

##### Calcium

As shown in Table 1, salinity significantly reduced the calcium content in the seedlings from the unprimed seeds of both barley species. This reduction was more prominent when the seeds were treated with 200 mM NaCl. The calcium concentration decreased by 58% and 57.9%in the coleoptile of *H. vulgare* (L. Manel) and *H. maritimum*, respectively, and by 62% and 21% in the radicle of *H. vulgare* (L. Manel) and *H. maritimum*, respectively, in comparison with their untreated control. Priming with CaCl_2_, KCl, and KNO_3_ increased the concentration of this nutrient in germinated seeds with all salt treatments. For *H. vulgare* (L. Manel), CaCl_2_ was the most powerful agent by contributing to the calcium concentration enhancement by 119% and 135%in the coleoptile and radicle, respectively, of the seedlings subjected to 200 mM NaCl, compared with their non-primed stressed equivalents. In addition, for *H. maritimum*, CaCl_2_ induced the greatest calcium content of coleoptile by 90% when the seeds were exposed to 200 mM NaCl and of radicle by 32% when the seeds were subjected to 100 mM NaCl when compared with the unprimed NaCl treated seeds.

##### Magnesium

Under unprimed conditions, the values of the magnesium content in the seedling were clearly decreased by both NaCl treatments (Table 1). The most important decline was noted in the seedlings stressed with 200mMfor both species. Indeed, it reached 53% and 50.4% in coleoptile, and 53.8% and 47.9% in radicle in *H. vulgare* (L. Manel) and *H. maritimum*, respectively, compared with the untreated control. Nevertheless, under salinity stress, the magnesium concentration was higher in the seedlings primed with different agents than in the unprimed stressed controls. Interestingly, CaCl_2_ seemed to be the most powerful one to remove the depressive effect of salt and to procure a greater improvement in the magnesium content, especially in *H. vulgare* (L. Manel). The rate of stimulation was 127% and 102% in the coleoptile and radicle, respectively, of seedlings subjected to 200 mM NaCl, compared with their stressed unprimed equivalents. Regarding *H. maritimum* species, in coleoptile, the increase in magnesium content was higher when the primed seeds with CaCl_2_ were exposed to 200 mM NaCl (74%). However, the best stimulation in radicle belonged to primed seeds with KNO_3_ subjected to 100 mM NaCl (31%) in comparison with their stressed unprimed equivalents.

##### Potassium

The potassium content was affected by salt stress in the absence of a priming treatment (Table 2). The 200 mM NaCl treatment reduced the content of this nutrient by 50% and 33.8% in coleoptile, and by 80.7% and 25% in radicle, in *H. vulgare* (L. Manel) and *H. maritimum,* respectively, as compared with their untreated controls. However, seed priming with KCl, KNO_3_, or CaCl_2_ had an effective role at improving the potassium concentration in the seedlings from both species subjected to either 100 mM or 200 mM NaCl. In fact, under the effect of 200 mM, this improvement was more important in the coleoptile of seedling from seeds primed with KNO_3_ (46.2%) and in the radicle of those primed with CaCl_2_ (33%) compared with the seedling from unprimed stressed equivalents of *H. vulgare* (L. Manel). Regarding *H. maritimum* species, the best rate of enhancement belonged to the seedlings primed with CaCl_2_ subjected to 200 mM NaCl. They reached 25.7% and 27.7% in the coleoptile and radicle, respectively, in comparison with the stressed unprimed equivalents (Table 2).

##### Sodium

It is noteworthy that, for unprimed seedlings, salt stress stimulated the absorption of sodium in both species (Table 2).As apparent from the measured values, the accumulation of sodium ions was remarkably higher when the seeds of *H. vulgare* (L. Manel) and *H. maritimum* were subjected to 200 mM NaCl compared with the untreated control. Interestingly, seed priming with several agents significantly decreased the sodium content in the seeds stressed with different salt treatments. As shown in Table 2, CaCl_2_ and KNO_3_ caused similar reduction rates for the sodium concentration, especially when the seeds were subjected to 100 mM NaCl, as compared with their stressed unprimed equivalent controls (Table 2).

##### Sodium/Potassium (Na/K) Ratio

Because of the significant increase in sodium content and decrease in potassium concentration, the salinity increased the sodium/potassium ratio in the seedlings of both species (Table 2). However, seed priming with all agents decreased it. Nevertheless, CaCl_2_ and KNO_3_ seemed to be powerful agents for maintaining the potassium content and limiting sodium absorption. Thus, they decreased the sodium/potassium ratio in the primed barley seeds germinated with several salt doses compared with the unprimed ones. Their improvement was markedly noted in *H. vulgare* (L. Manel) submitted to 100 mM NaCl, compared with their stressed controls (Table 2).

### 2.3. Effect of Seed Priming on Seedling Electrolyte Leakage

As shown in Figure 5, under unprimed conditions, both salt concentrations significantly increased electrolyte leakage in the seedlings of both barley species. The 200 mM NaCl treatment increased the membrane stability index (MSI) levels by 110% and 30% in the radicle and by 56% and 45% in the coleoptile for *H. vulgare* (L. Manel) and *H. maritimum,* respectively, as compared with the control. Interestingly, we noted that the MSI levels were lower in the seedlings from the seeds primed with all agents than in the unprimed controls under both salt treatments. This beneficial effect was more efficient when we applied seed priming with CaCl_2_ or KNO_3_. Both agents were able to provide a similar decrease in the radicle and coleoptile electrolyte leakage of *H. vulgare* (L. Manel) and *H. maritimum* species subjected to either 100 mM or 200 mM NaCl (Figure 5).

### 2.4. Effect of Seed Priming onLipid Peroxidation

The results demonstrated that the exposure of unprimed seeds of both barley species to different salt treatments exhibited a significant increase in malondialdehyde (MDA) levels when compared with the control (Figure 6). This increase was observed when the seeds were subjected to 200 mM NaCl. This salt concentration promoted MDA accumulation by 68.1% and 79.6% in the coleoptile and radicle, respectively, of *H. maritimum* seedlings compared with their controls. Nevertheless, this stimulation was even more prominent in *H. vulgare* (L. Manel), with an increase of about 160.3% and 197.6% in comparison with their controls in the coleoptile and radicle, respectively. Seedlings derived from primed seeds were less affected by salinity. In particular, in *H. vulgare* (L. Manel), CaCl_2_ seemed to be the best compound, playing a crucial role in alleviating salinity’s harmful effect. It reduced MDA values by 63% and 72% in the coleoptile and radicle, respectively, when seeds were subjected to 200 mM NaCl compared with the control. Regarding *H. maritimum* seeds exposed to 200 mM NaCl, a similar significant decrease in the MDA content was recorded in the seedlings from primed seeds with CaCl_2_ and those soaked in KNO_3_ compared with its content in the seedlings from unprimed stressed seeds (Figure 6).

### 2.5. Pearson’s Correlation Matrix Analysis

Pearson’s correlation matrix considered all traits including the FG%, growth parameters (length and dry weight), mineral elements analysis, membrane stability index (MSI), and MDA content for both barley species *H. vulgare* (L. Manel) (Table 3) and *H. maritimum* (Table 4). The results shown for the correlations matrix are concomitant with those obtained from the trait-by-trait analyses. In fact, we noticed a significant effect of all parameters measured in the unprimed seeds of *H. vulgare* (L. Manel) species subjected to 200 mM NaCl treatment. Under these restrictive conditions, the results showed that this salt concentration was negatively correlated with FG%, length, and dry weight, especially in the coleoptile length (Pearson’s correlation coefficient r = −0.615) and radicle dry weight (r = −0.6708). The results also demonstrated a great reduction in essential nutrient content, notably in the coleoptile potassium content (r = −0.809), iron concentration (r = −0.730), and magnesium concentration (r = −0.735). In contrast, a positive correlation for this salt treatment was markedly registered for sodium content, sodium/potassium ratio, MSI level, and MDA content (Table 3). In accordance with the trait-by-trait analyses results, under the effect of 100 mM NaCl, CaCl_2_ seed priming appeared to be the best agent, and was positively correlated with the majority of the parameters studied versus the remaining priming treatments (KCl or KNO_3_) (Table 3). Indeed, its effect was more significant in the radicle and coleoptile dry weight (r = 0.507 and 0.570, respectively), iron content (r = 0.262), and calcium concentration in the radicle (r = 0.313). However, it was negatively correlated with the sodium concentration and sodium/potassium ratio, and particularly with the coleoptile MSI (r = −0.446) and coleoptile MDA content (r = −0.396). In addition, for *H. maritimum*, the results shown by the correlations matrix demonstrated a prominent effect of 200 mM NaCl treatment in unprimed seedlings. We noticed a great reduction in FG%, dry weight, length, and essential element content against an increase of MDA, MSI, and sodium concentrations, as well as a stimulation of the sodium/potassium ratio (Table 4). In agreement with the trait-by-trait analyses of the results, treatment with 100 mM NaCl + CaCl_2_ appeared to be positively correlated with the majority of the parameters studied (Table 4).

## 3. Discussion

Successful and rapid seed germination is critical factor affecting crop production. These stages are the most sensitive to abiotic stress, especially salt stress [7]. As shown in our results, *Hordeum vulgare* (L. Manel) and *Hordeum maritimum* seedlings from unprimed seeds exhibited classical responses to the deleterious effects of salinity. While salt stress affected the development of all seedlings from unprimed seeds, the domesticated glycophyte species suffered more than the native halophyte one. In fact, the dry weight of the radicle and coleoptile was reduced under the effect of 200 mM NaCl. In addition, salt initiated a great decrease in the radicle and coleoptile lenght. Our study is in agreement with previous studies, showing that glycophytes are sensitive to salinity during germination and seedling growth, and that they could not survive under high salinity conditions [39]. In fact, salinity reduces major glycophyte crop germination, growth, and production, such as corn, wheat, soybean, sorghum, oat, barley, potato, and sugar beet [40]. This effect is caused by an osmotic stress, by which salinity increases the external osmotic potential that limits the seedlings’ capacity for water absorption and thus inhibits its cell growth [41]. In addition, salinity may affect the germination of seeds through the toxic effects of an excess amount of sodium and chloride ions on embryo viability [42,43]. Thus, toxic ion accumulation reduces the efficiency of seed reserve mobilization during germination and blocks the transport of essential nutrient [44]. Indeed, as mentioned above, salt treatment remarkably decreased the essential nutrient content such as the concentration of potassium and magnesium in coleoptile, in addition to the radicle calcium concentration. In contrast, it stimulated the sodium/potassium ratio in the radicle and coleoptile. A high amount of accumulated sodium and decreased calcium level in NaCl-treated seedlings may be caused by sodium replacement of calcium in the cell walls and vacuoles [45,46]. Moreover, despite potassium being generally regarded as a main competitor of sodium in uptake and transport, the content of this element decreased in stressed seedlings, which induced potassium leakage and competition with sodium [5]. Furthermore, seedlings from the seeds treated with 200 mM NaCl had a higher level of electrolyte leakage (MSI) and malondialdehyde content (MDA) in the radicle and coleoptile compared with the untreated seeds. This increase is a sign of membrane damage at a cellular level under salt stress and serves as an important oxidative stress indicator resulting in the inhibition of seed germination [47,48]. In addition, many previous studies have indicated that salt stress induces the production of reactive oxygen derivatives (ROS) resulting in the peroxidation of mitochondrial lipids, loss of membrane integrity, and degradation of proteins and inactivation of enzymes [47]. Data have shown that *H. maritimum* species suffered less decline in potassium concentration compared with *H. vulgare* (L. Manel). Therefore, the sodium/potassium ratio in *H. maritimum* is lower than that observed in *H. vulgare* seedlings. These results further support the idea that the salt tolerance of *H. maritimum* may be based on maintaining the seedling sodium/potassium balance [49,50]. Interestingly, the higher degree of salt tolerance of *H. maritimum* compared with *H. vulgare* was coupled to its higher competence to preserve the membrane integrity. In fact, our results showed that the seedlings of the native species occurred at lower values of MDA and electrolyte leakage than the cultivated ones. Previous investigation by Seckin et al. proved that the MDA content and electrolyte leakage changed in *H. maritimum* only at a high salt concentration (200 and 300 mM NaCl); furthermore, in *H. vulgare,* these two parameters decreased even under low salinity levels, because of oxidative damage [51]. At the present time, seed priming is a low cost and low environment risk technique used to enhance growth under both favorable and unfavorable conditions [52]. Additionally, this approach is helpful to overcome salinity problems in the agricultural area by alleviating dormancy and promoting seed germination and seedling establishment under stressful conditions [53]. Furthermore, it provides faster and more synchronized germination of a great number of crops, such as tomato [14], without deteriorating natural resources. The overall results proved that, although the effects of different priming agents differed, all of them had the capacity to alleviate the stressful impact of salinity, and promoted the germination of barley seeds under salt stress with a little extent of difference. Yet, it is imperative to mention that this improvement occurs in a salt concentration-, priming agent-, and species-dependent manner. In this context, it was proven that seed priming is able to reduce the physical resistance of the endosperm during imbibition, to repair membranes, and to develop immature embryos [54]. The results of our experiment were consistent with several research works in this area. Ruan et al. showed that seed priming with KCl and CaCl_2_ enhanced several germination parameters in rice via induction of the mitotic activity and limitation of the absorption of sodium and chloride ions [26]. Similarly, Iqbal et al. reported that seed priming with KCl and CaCl_2_ alleviated salt stress damage in maize, thereby increasing the shoot and dry fresh weight [55]. In addition, Nasri et al. revealed that lettuce seeds soaked with KNO_3_ or CaCl_2_ and germinated under salt stress generally increased calcium metabolism and transport in tissues [13]. Additionally, priming with a potassium nitrate solution showed advantageous effects on the germination and growth rate of a wide range of crops under stressful environments, such as tomato [56] and pepper [54]. In addition, investigation in the impact of seed priming on some germination aspects in canola showed that seed priming with KCl 2% increased the root dry weight and mean germination rate at suitable priming times, and enhanced faster seedling establishment and improved plant tolerance against adverse environmental conditions [57].

The effective impact of CaCl_2_ pretreatment in barley seeds observed in the current work could be explained by the fact that calcium is known to be one of the most important essential nutrients necessary for plant growth and stress tolerance enhancement [58]. During germination, calcium ions play an important role in the regulation of plant metabolism, and maintain the intracellular potassium and sodium ion balance, which lead to an alleviation of the toxic sodium ion effects and to an improvement of salt tolerance [59]. In addition, when plants are exposed to NaCl treatments, the concentration of intracellular calcium ions increases rapidly, and Köster et al. proved that calcium ions coordinate the response to salt stress at cellular levels [60]. In this context, Jafar et al. reported that although different agents of priming improved the tolerance of wheat to salinity, CaCl_2_ was the most effective compound compared with ascorbic acid, salicylic acid, and kinetin [61]. However, potassium ions are not able to play a similar physiological role as the second messengers [62]. Taken together, these beneficial effects of calcium ions make calcium chloride priming the best treatment for barley seed germination under salt stress. Furthermore, rapid hydration may cause the leakage of essential nutrients out of the seed during germination, resulting in seed damage in some species at higher stress treatments [63].

## 4. Materials and Methods

### 4.1. Plant Material and Growth Conditions

Two barley species, *H*. *maritimum* (wild) and *H.vulgare* (L. Manel) (cultivated), were used throughout the experiment. *Hordeum maritimum* seeds (=H. marinium Huds. subs. Marinum, 2n + 14) were collected from Kalbia Sebkha (an intermittent in Tunisia that covers 8000 hectares in Sousse governorate at 35°’50′34” North, 10°’16′18” East South of Kondar); however, *Hordeum vulgare* (L. Manel) seeds were provided by the National Institute of Agronomy of Tunis (INAT). The study was conducted in the Extremophile Plants Laboratory (LPE) of the Biotechnology Center of Borj Cedria (CBBC), Tunisia. Uniform barley seeds were selected and disinfected for 8 min with sodium hypochlorite (2%, *v/v*), then, they were abundantly rinsed with distilled water. Then, the seeds were soaked in distilled water (unprimed seeds), primed with 5 mM of CaCl_2_ solution for 20 h, or soaked with 270 mM KNO_3_ or 200 mM KCl for 40 h. The concentrations and times of the priming agents were chosen based on our own preliminary test, later confirmed by a comparison with previous investigations such, as in sorghum [64] and rice [65]. Finally, the seeds were placed in 20 Petri dishes moistened with a double layer of ashless filter paper*with 25 seeds for each treatment. The Petri dishes were moistened with three different saline concentrations consisting of 0, 100, and 200 mM NaCl, and were placed in a germinator at 20 °C, in the dark, for seven days.

### 4.2. Germination Parameters

#### 4.2.1. Final Germination Percentage (FG %)

FG% was calculated from 20 repetitions of each treatment, as described by Nasri et al., based on the following equation [13]:

FG% = (total germinated seeds/total number of seeds) * 100

#### 4.2.2. Mean Daily Germination (MDG) 

According to Osborne (1993): MDG = Germination rate/Xth test day [66].

### 4.3. Seedlings Growth

At the end of the experiment, 10 seedlings from each treatment were harvested and rinsed carefully in many baths of distilled water. For seedling growth, independent length and dry matter measurements were performed on separated coleoptiles and radicles. The length was measured with a ruler and the dry weight (DW) was determined after desiccation at 80 °C for 72 h in an oven using a precision balance (Mettler type AE100 at 1/100 of mg).

### 4.4. Nutrient Extraction and Analysis

First, 20 mg of dried plant material was digested with 10 mL of sulfuric acid (H_2_SO_4_, 1N) for 1 h at 80 °C, and then left overnight at room temperature (five repetitions for each treatment).This allowed for the total extraction of the target elements of the samples. The iron, magnesium, and calcium were analyzed using an atomic absorption spectrophotometer (Perkin Elmer, Analyst 300, Rodgau, Germany). However, the potassium and sodium ions were determined using a flame photometer (BWB Technologies XP, France).The results were expressed in μg·g^−1^ DW for and in mg·g^−1^ DW for the magnesium, calcium, potassium, and sodium, according to the method illustrated by Farhat et al. [67].

### 4.5. Electrolyte Leakage

Electrolyte leakage was estimated by measuring the membrane stability index (MSI) using a digital conductivity meter (BANTE instruments 950, China). Ten repetitions were carried on for each treatment. Initially, we cut 0.2 g of fresh material and placed it in falcon tubes filled with 10 mL of distilled water and heated it at 32 °C for 30 min, and finally, we recorded the electrical conductivity (EC1). The conductivity (EC2) was also assayed after placing the samples at 100 °C for 2 h. The MSI was calculated according to Dionioese and Tobita, using the following formula Equation (1) [68].
(1)$$\mathrm{MSI}\;=\left(\frac{\mathrm{EC}1}{\mathrm{EC}2}\right)*100$$

EC1: electrical conductivity measured at the first time

EC2: electrical conductivity measured at the second time

### 4.6. Malondialdehyde Content (MDA)

The level of lipid peroxidation was measured as the 2-thiobarbituric acid-reactive substances (mainly malondialdehyde) according to Bueg and Aust [69]. Frozen samples (1 g of fresh material, five repetitions for each treatment) were homogenized with a pre-chilled mortar and pestle with 10 mL of (0, 1%; p/v) trichloroacetic acid, and were centrifuged at 10,000 g for 10 min and at 25 °C. Then, 1mL of the upper liquid layer (supernatant) was added to 4 mL thiobarbituric acid (TBA) (0, 5 %; p/v). After centrifugation at 1000 g for 10 min, the supernatant absorbance was read (532 nm) and the values corresponding to non-specific absorption (600 nm) were subtracted. The MDA concentration was calculated using its molar extinction coefficient (155 mM^−1^·cm^−1^) according to the following formula Equation (2):(2)$$MDA\;\left(nm/gFW\right)=\left(\frac{\left(DO532-DO600\right)\;*VS}{0.155}\right)*FW\;$$

### 4.7. Statistical Analysis

Statistical analyses were performed with the “XLSTAT” software (version 2014). The mean values and standard error (SE) were obtained from at least 20 replicates for the factors of germination (FG%, MDG); 10 replicates for length, dry weight, and electrolyte leakage (MSI); and from 5 replicates for the ion content and MDA, and were analyzed using Duncan’s multiple range test. A p value less than 0.05 was considered statistically significant. Pearson’s correlation analysis was done based on principal component analysis (PCA) using XLSTAT software, considering the variables centered on their means and normalized with a standard deviation of 1.

*: One of our objectives was to analyze the ion content, so we used a specific paper called ashless filter paper. This type of paper does not interfere with the germination process and does not contain elements that could be absorbed by the seeds.

## 5. Conclusions

This work led us to conclude that salinity stress decreased the germination parameters and seedling growth of both barley species, and were associated with a significant reduction of essential nutrient mobilization, an intensive increase of sodium content, and a stimulation of electrolyte leakage and MDA levels. Hazardous impacts of salinity were more pronounced for the *H. vulgare* (L. Manel) species. *H. maritimum* was able to maintain its germinative and physiological parameters by developing different defense mechanisms. Under salinity, different priming treatments promoted barley germination and seedling growth, and provoked a greater stress tolerance of primed seeds to different extents. Principal component analysis and Pearson’s correlation analysis clearly indicated that CaCl_2_ priming was better than the other priming agents due to its positive effects on the germination and growth traits. Bearing in mind that primed seeds with several agents of both barley species may respond differently to salt treatments, it will be necessary to continue this work at a later developmental stage. Further research is necessary to determine the ability of these agents to promote plant growth under permanent salt stress conditions.

## Figures and Tables

**Figure 1 plants-10-02264-f001:**
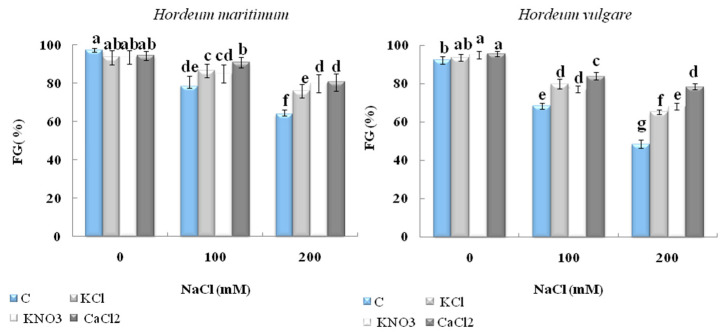
Effect of KCl, KNO_3_, and CaCl_2_ seed priming on the final germination rate (FG%) of two barley species, *Hordeum maritimum* and *Hordeum vulgare* (L. Manel),subjected to various salt concentrations (0, 100, and 200 mM NaCl) for 7 days at the germination stage. Values are means of 20 replicates ± standard error. Data with the same letter are not significantly different at *p* < 0.05 (Duncan’s test).

**Figure 2 plants-10-02264-f002:**
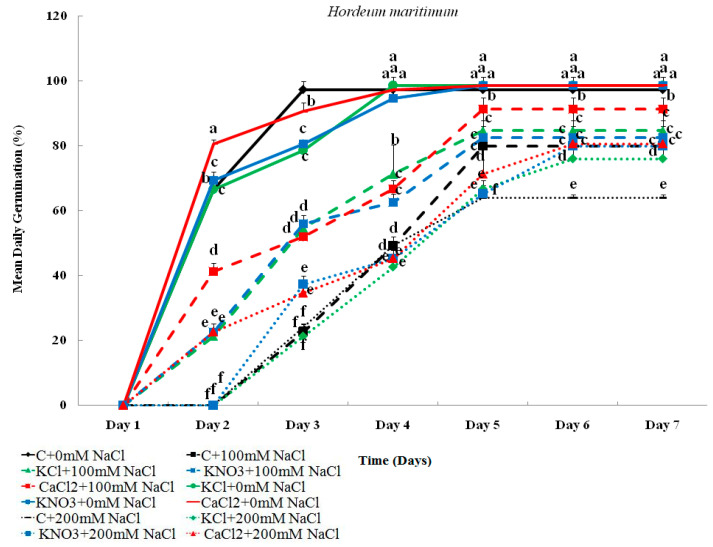
Effect of KCl, KNO_3_, and CaCl_2_ seed priming on the mean daily germination (MDG) in *Hordeum maritimum* and *Hordeum vulgare* (L. Manel) subjected to various salt concentrations (0, 100, and 200 mM NaCl) for 7 days at the germination stage. Values are means of 20 replicates ± standard error. Data with the same letter are not significantly different at *p* < 0.05 (Duncan’s test).

**Figure 3 plants-10-02264-f003:**
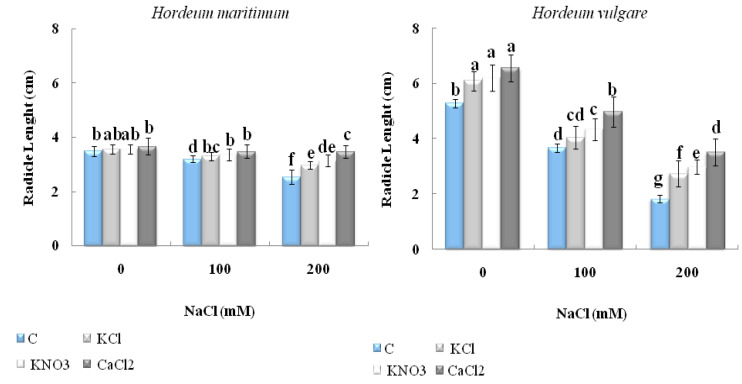
Effect of KCl, KNO_3_, and CaCl_2_ seed priming on the radicle length and coleoptile length of two barley species, *Hordeum maritimum* and *Hordeum vulgare* (L. Manel), subjected to various salt concentrations (0, 100, and 200 mM NaCl) for 7 days at the germination stage. Values are means of 20 replicates ± standard error. Data with the same letter are not significantly different at *p* < 0.05 (Duncan’s test).

**Figure 4 plants-10-02264-f004:**
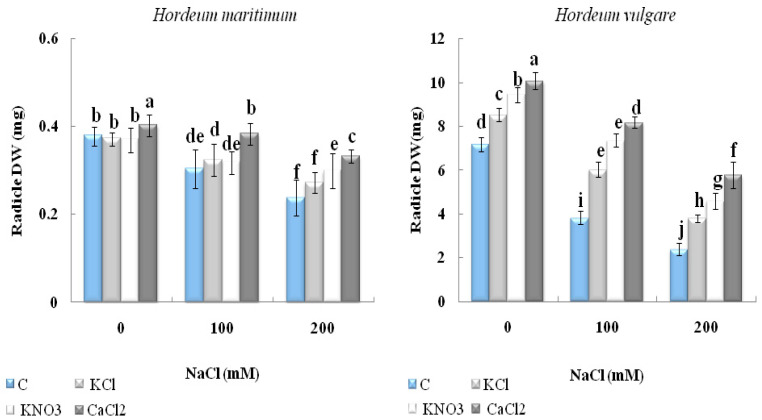
Effect of KCl, KNO_3_, and CaCl_2_ seed priming on the dry weight (DW) on the radicle and coleoptile of two barley species, *Hordeum maritimum* and *Hordeum vulgare* (L. Manel)*,* subjected to various salt concentrations (0, 100, and 200 mM NaCl) for 7 days at the germination stage. Values are the means of 10 independent replicates ± standard error. Data with the same letter are not significantly different at *p* < 0.05 (Duncan’s test).

**Figure 5 plants-10-02264-f005:**
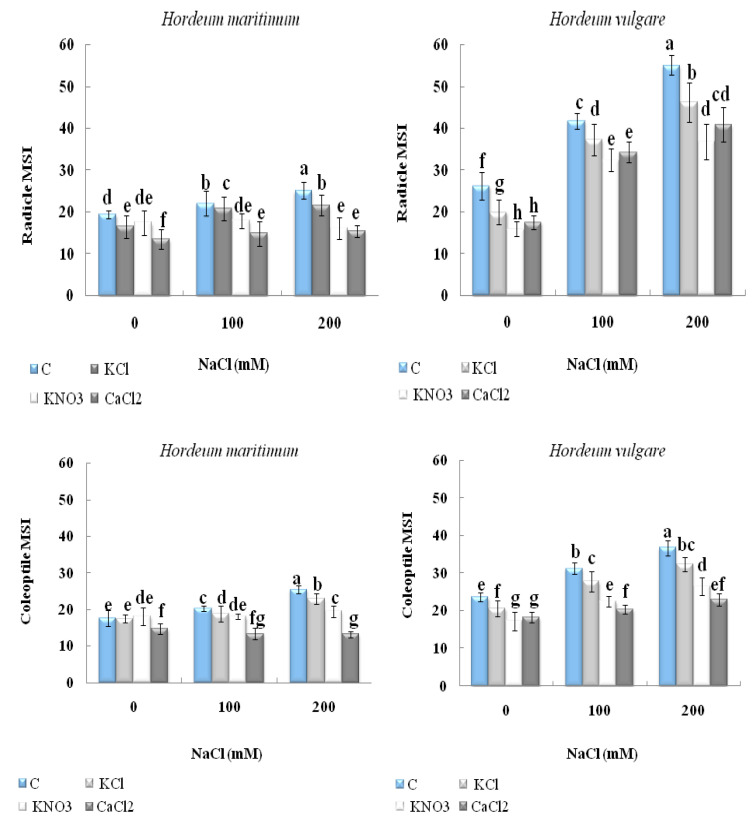
Effect of KCl, KNO_3_, and CaCl_2_ seed priming on the membrane stability index (MSI) on the radicle and coleoptile of two barley species, *Hordeum maritimum* and *Hordeum vulgare* (L. Manel), subjected to various salt concentrations (0, 100, and 200 m MNaCl) for 7 days in the germination stage. Values are the means of 5 independent replicates ± standard error. Data with the same letter are not significantly different at *p* < 0.05 (Duncan’s test).

**Figure 6 plants-10-02264-f006:**
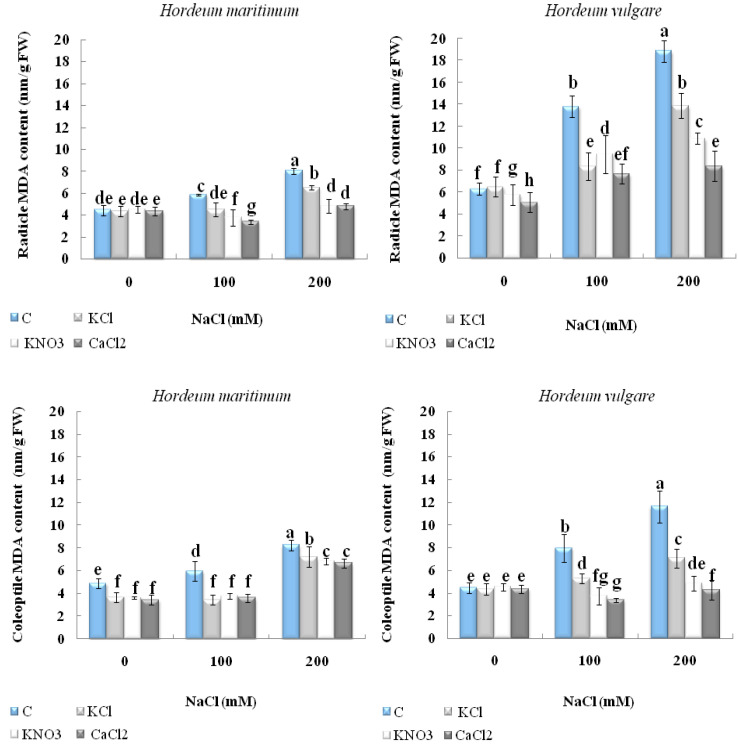
Effect of KCl, KNO_3_, and CaCl_2_ seed priming on the malondialdehyde content (MDA) on the radicle and coleoptile of two barley species, *Hordeum maritimum* and *Hordeum vulgare* (L. Manel), subjected to various salt concentrations (0, 100, and 200 mM NaCl) for 7 days at the germination stage. Values are means of 5 independent replicates ± standard error. Data with the same letter are not significantly different at *p* < 0.05 (Duncan’s test).

**Table 1 plants-10-02264-t001:** Effect of KCl, KNO_3_, and CaCl_2_ seed priming on the total Fe, Ca, and Mg content on the radicle (R) and coleoptile (Co) of two barley species, *Hordeum maritimum* and *Hordeum vulgare* (L. Manel), subjected to various salt concentrations (0, 100, and 200 mM NaCl) for 7 days at the germination stage.

Treatment (mM NaCl)	C (Unprimed Seeds) + 0	C (Unprimed Seeds) + 100	C (Unprimed Seeds) + 200	KCl + 100	KCl + 200	KNO_3_ + 100	KNO_3_ + 200	CaCl_2_ + 100	CaCl_2_ + 200
*H.vulgare* (L. Manel)									
Fe Co (mg/gDW)	4994 ± 29 ^a^	2639 ± 141 ^c^	1414 ± 213 ^e^	3009 ± 331 ^bc^	2235 ± 126 ^d^	2974 ± 331 ^bc^	2540 ± 309 ^cd^	3359 ± 249 ^b^	2679 ± 357 ^c^
Fe R (mg/gDW)	2920 ± 45 ^a^	1919 ± 161 ^d^	1276 ± 90 ^f^	2688 ± 199 ^b^	1784 ± 102 ^e^	2524 ± 202 ^b^	2116 ± 79 ^cd^	2912 ± 128 ^a^	2292 ± 178 ^c^
Ca Co (mg/gDW)	18.25 ± 1.2 ^a^	11.68 ± 0.59 ^d^	7.60 ± 0.59 ^e^	14.91 ± 0.7 ^c^	11.04 ± 0.55 ^d^	13.63 ± 0.79 ^c^	11.10 ± 1.24 ^d^	19.75 ± 1.03 ^a^	16.66 ± 1.73 ^b^
Ca R (mg/gDW)	14.93 ± 0.9 ^c^	8.84 ± 1.14 ^f^	5.66 ± 1.14 ^g^	11.3 ± 0.74 ^de^	10.21 ± 0.76 ^e^	12.8 ± 1.36 ^cd^	1146 ± 0.9 ^de^	17.94 ± 0.78 ^a^	13.32 ± 1.26 ^b^
Mg Co (mg/gDW)	22.73 ± 1.5 ^a^	16.19 ± 1.11 ^d^	10.65 ± 1.11 ^e^	20.51 ± 1.14 ^b^	18.73 ± 1.37 ^bc^	20.16 ± 1.6 ^bc^	18.15 ± 0.68 ^c^	24.09 ± 1.68 ^a^	24.26 ± 1.7 ^a^
Mg R (mg/gDW)	25.15 ± 1.33 ^a^	17.58 ± 0.74 ^c^	11.62 ± 0.74 ^d^	23.5 ± 1.98 ^ab^	18.96 ± 0.99 ^c^	22.17 ± 2.36 ^b^	17.33 ± 0.4 ^c^	25.43 ± 2.26 ^a^	23.5 ± 1.99 ^ab^
*H. maritimum*									
Fe Co (mg/gDW)	1178 ± 34 ^a^	851 ± 26 ^e^	725 ± 19.04 ^f^	1051 ± 43 ^b^	896 ± 50 ^de^	1041 ± 41 ^b^	953 ± 18 ^c^	1022 ± 37 ^b^	918 ± 35 ^cd^
Fe R (mg/gDW)	992 ± 24 ^b^	921 ± 43 ^c^	646 ± 33.92 ^f^	916 ± 60 ^c^	737 ± 38 ^e^	982 ± 45 ^bc^	838 ± 31 ^d^	1055 ± 42 ^a^	936 ± 44 ^bc^
Ca Co (mg/gDW)	0.887 ± 0.049 ^a^	0.579 ± 0.06 ^cd^	0.373 ± 0.057 ^e^	0.639 ± 0.055 ^bc^	0.526 ± 0.060 ^d^	0.605 ± 0.055 ^c^	0.525 ± 0.031 ^d^	0.855 ± 0.042 ^a^	0.711 ± 0.035 ^b^
Ca R (mg/gDW)	0.391 ± 0.023 ^ab^	0.314 ± 0.030 ^c^	0.181 ± 0.03 ^e^	0.323 ± 0.031 ^c^	0.199 ± 0.026 ^e^	0.359 ± 0.015 ^b^	0.194 ± 0.013 ^e^	0.417 ± 0.025 ^a^	0.238 ± 0.023 ^d^
Mg Co (mg/gDW)	0.228 ± 0.026 ^a^	0.187 ± 0.017 ^c^	0.115 ± 0.017 ^e^	0.186 ± 0.024 ^c^	0.141 ± 0.009 ^de^	0.158 ± 0.017 ^d^	0.166 ± 0.016 ^d^	0.227 ± 0.016 ^ab^	0.2 ± 0.018 ^b^
Mg R (mg/gDW)	0.217 ± 0.024 ^a^	0.166 ± 0.009 ^c^	0.113 ± 0.009 ^f^	0.186 ± 0.013 ^b^	0.149 ± 0.013 ^d^	0.218 ± 0.017 ^a^	0.134 ± 0.008 ^e^	0.202 ± 0.017 ^ab^	0.146 ± 0.017 ^d^

Values are means of 5 independent replicates ± standard error. Data with the same letter are not significantly different at *p* < 0.05 (Duncan’s test).

**Table 2 plants-10-02264-t002:** Effect of KCl, KNO_3_, and CaCl_2_ seed priming on the Na and K content and Na/K ratio on the radicle (R) and coleoptile (Co) of two barley species, *Hordeum maritimum* and *Hordeum vulgare* (L. Manel), subjected to various salt concentrations (0, 100, and 200 mM NaCl) for7 days at the germination stage.

Treatment (mM NaCl)	C (Unprimed Seeds) + 0	C (Unprimed Seeds) + 100	C (Unprimed Seeds) + 200	KCl + 100	KCl + 200	KNO_3_ + 100	KNO_3_ + 200	CaCl_2_ + 100	CaCl_2_ + 200
*H.vulgare* (L. Manel)									
Na Co (mg/gDW)	0.066 ± 0.005 ^g^	0.879 ± 0.049 ^d^	1.63 ± 0.042 ^a^	0.67 ± 0.06 ^e^	1.262 ± 0.043 ^b^	0.617 ± 0.091 ^ef^	0.912 ± 0.052 ^d^	0.566 ± 0.051 ^f^	1.03 ± 0.07 ^c^
Na R (mg/gDW)	0.08 ± 0.011 ^h^	0.539 ± 0.023 ^d^	1.01 ± 0.023 ^a^	0.458 ± 0.037 ^b^	0.744 ± 0.031 ^b^	0.371 ± 0.044 ^f^	0.769 ± 0.05 ^b^	0.318 ± 0.033 ^g^	0.67 ± 0.027 ^c^
K Co (mg/gDW)	0.9 ± 0.036 ^a^	0.65 ± 0.02 ^c^	0.45 ± 0.013 ^e^	0.67 ± 0.03 ^bc^	0.599 ± 0.038 ^d^	0.719 ± 0.041 ^b^	0.658 ± 0.03 ^c^	0.695 ± 0.021 ^bc^	0.6 ± 0.016 ^d^
K R (mg/gDW)	0.778 ± 0.048 ^a^	0.365 ± 0.021 ^c^	0.15 ± 0.014 ^f^	0.391 ± 0.018 ^c^	0.2 ± 0.019 ^e^	0.453 ± 0.03 ^b^	0.228 ± 0.021 ^de^	0.403 ± 0.035 ^c^	0.26 ± 0.01 ^d^
Na/K Co(mg/gDW)	0.073 ± 0.007 ^g^	1.34 ± 0.116 ^d^	3.55 ± 0.058 ^a^	0.98 ± 0.089 ^e^	1.53 ± 0.153 ^c^	0.86 ± 0.142 ^ef^	1.924 ± 0.153 ^b^	0.814 ± 0.072 ^f^	1.29 ± 0.1 ^d^
Na/K R (mg/gDW)	0.054 ± 0.044 ^f^	1.46 ± 0.157 ^d^	6.739 ± 0.587 ^a^	1.16 ± 0.06 ^de^	3.75 ± 0.321 ^b^	0.824 ± 0.114 ^e^	3.414 ± 0.443 ^b^	0.807 ± 0.117 ^e^	2.573 ± 0.2 ^c^
*H. maritimum*									
Na Co (mg/gDW)	0.114 ± 0.015 ^g^	1.397 ± 0.1 ^c^	1.936 ± 0.059 ^a^	1.091 ± 0.083 ^e^	1.568 ± 0.77 ^b^	0.935 ± 0.086 ^f^	1.418 ± 0.084 ^c^	0.959 ± 0.076 ^f^	1.24 ± 0.056 ^d^
Na R (mg/gDW)	0.134 ± 0.011 ^e^	0.942 ± 0.058 ^bc^	1.134 ± 0.058 ^a^	0.91 ± 0.044 ^c^	1.005 ± 0.029 ^b^	0.876 ± 0.066 ^d^	0.957 ± 0.029 ^bc^	0.871 ± 0.048 ^d^	0.87 ± 0.05 ^d^
K Co (mg/gDW)	1.296 ± 0.056 ^a^	1.191 ± 0.03 ^b^	0.871 ± 0.067 ^e^	1.206 ± 0.02 ^b^	1.009 ± 0.06 ^d^	1.24 ± 0.06 ^ab^	0.936 ± 0.069 ^de^	1.213 ± 0.034 ^b^	1.095 ± 0.02 ^c^
K R (mg/gDW)	0.963 ± 0.33 ^ab^	0.89 ± 0.034 ^bc^	0.720 ± 0.04 ^e^	0.966 ± 0.03 ^ab^	0.84 ± 0.059 ^de^	0.87 ± 0.049 ^c^	0.761 ± 0.058 ^e^	0.989 ± 0.045 ^a^	0.92 ± 0.03 ^b^
Na/K Co(mg/gDW)	0.088 ± 0.013 ^f^	1.173 ± 0.08 ^c^	2.265 ± 0.169 ^a^	0.90 ± 0.066 ^d^	1.561 ± 0.145 ^b^	0.751 ± 0.067 ^e^	1.520 ± 0.099 ^b^	0.792 ± 0.08 ^de^	1.13 ± 0.067 ^c^
Na/K R (mg/gDW)	0.217 ± 0.024 ^e^	0.166 ± 0.009 ^c^	0.113 ± 0.009 ^a^	0.186 ± 0.013 ^cd^	0.149 ± 0.013 ^b^	0.218 ± 0.017 ^cd^	0.134 ± 0.008 ^b^	0.202 ± 0.017 ^d^	0.146 ± 0.017 ^cd^

Values are the means of 5 independent replicates ± standard error. Data with the same letter are not significantly different at *p* < 0.05 (Duncan’s test).

**Table 3 plants-10-02264-t003:** *Hordeum vulgare* (L. Manel) Pearson’s correlation matrix: C L: coleoptile length; R L: radicle length; C DW: coleoptile dry weight; R DW: radicle dry weight; C MSI: coleoptile membrane stability index; R MSI: radicle membrane stability index; C MDA: coleoptile malondialdehyde content; R MDA: radicle malondialdehyde content; C Ca: coleoptile calcium content; R Ca: radicle calcium content; C Mg: coleoptile magnesium content; R Mg: radicle magnesium content; C K: coleoptile potassium content; R K: radicle potassium content; C Fe: coleoptile iron content; R Fe: radicle iron content; C Na: coleoptile sodium content; R Na: radicle sodium content; C Na/K: coleoptile sodium/potassium ratio; R Na/K: radicle sodium/potassium ratio; FG %: final germination rate.

Treatments		100 mM NaCl	200 mM NaCl	100 mM NaCl + KCl	200 mM NaCl + KCl	100 mM NaCl + KNO_3_	200 mM NaCl + KNO_3_	100 mM NaCl + CaCl_2_	200 mM NaCl + CaCl_2_
	Parameter								
C L		−0.172	**−0.615**	0.099	−0.222	**0.351**	−0.102	**0.472**	0.190
R L		0.058	**−0.537**	0.253	**−0.405**	0.146	−0.204	**0.554**	0.134
C DW		−0.298	**−0.565**	0.140	−0.302	**0.407**	−0.143	**0.570**	0.191
R DW		**−0.318**	**−0.670**	0.237	−0.236	0.308	−0.081	**0.507**	0.253
C MSI		0.270	**0.642**	0.038	0.274	**−0.313**	−0.049	**−0.446**	**−0.416**
R MSI		0.098	**0.708**	−0.130	**0.333**	**−0.383**	−0.155	−0.282	−0.189
C MDA		**0.328**	**0.766**	−0.108	0.154	**−0.313**	−0.152	**−0.396**	−0.278
R MDA		0.226	**0.751**	**−0.348**	0.254	−0.133	−0.022	**−0.409**	**−0.319**
C Ca		−0.164	**−0.580**	0.164	−0.230	0.034	−0.224	**0.657**	**0.342**
R Ca		−0.293	**−0.626**	−0.031	−0.150	0.120	−0.019	**0.657**	**0.342**
C Mg		−0.252	**−0.735**	0.123	−0.031	0.092	−0.081	**0.435**	**0.449**
R Mg		−0.200	**−0.691**	0.287	−0.086	0.177	−0.221	**0.446**	0.287
C Fe		0.020	**−0.730**	0.246	−0.227	0.225	−0.040	**0.461**	0.044
R Fe		−0.196	**−0.664**	**0.363**	−0.294	0.243	−0.052	**0.526**	0.074
C K		0.096	**−0.809**	0.202	−0.161	**0.392**	0.109	0.281	−0.112
R K		0.210	**−0.556**	0.301	**−0.381**	**0.522**	−0.281	**0.344**	−0.160
C Na		−0.074	**0.754**	−0.304	**0.347**	**−0.363**	−0.037	**−0.419**	0.098
R Na		−0.121	**0.684**	−0.260	0.229	**−0.408**	0.272	**−0.501**	0.105
C Na/K		−0.086	**0.901**	−0.246	−0.004	−0.303	0.172	**−0.324**	−0.109
R Na/K		−0.221	**0.814**	−0.281	0.227	**−0.347**	0.161	**−0.350**	−0.003
TG%		−0.065	**−0.733**	0.283	−0.239	0.196	−0.152	**0.457**	0.254

Variables arecentered around their means and are normalized with a standard deviation of 1, *n* = 5. Values in bold represent significant correlations at the 0.05 level.

**Table 4 plants-10-02264-t004:** *Hordeum maritimum* Pearson’s correlation matrix: C L: coleoptile length; R L: radicle length; C DW: coleoptile dry weight; R DW: radicle dry weight; C MSI: coleoptile membrane stability index; R MSI: radicle membrane stability index; C MDA: coleoptile malondialdehyde content; R MDA: radicle malondialdehyde content; C Ca: coleoptile calcium content; R Ca: radicle calcium content; C Mg: coleoptile magnesium content; R Mg: radicle magnesium content; C K: coleoptile potassium content; R K: radicle potassium content; C Fe: coleoptile iron content; R Fe: radicle iron content; C Na: coleoptile sodium content; R Na: radicle sodium content; C Na/K: coleoptile sodium/potassium ratio; R Na/K: radicle sodium/potassium ratio; FG %: final germination rate.

Treatments		100 mM NaCl	200 mM NaCl	100 mM NaCl + KCl	200 mM NaCl + KCl	100 mM NaCl + KNO_3_	200 mM NaCl + KNO_3_	100 mM NaCl + CaCl_2_	200 mM NaCl + CaCl_2_
	Parameter								
C L		−0.099	−0.291	0.107	−0.243	0.107	−0.291	**0.363**	**0.347**
R L		−0.006	**−0.779**	0.121	−0.190	0.195	0.140	0.287	0.232
C DW		0.064	**−0.552**	0.137	−0.234	0.192	−0.262	**0.534**	0.120
R DW		−0.059	**−0.410**	0.099	**−0.422**	0.057	−0.016	**0.591**	0.160
C MSI		0.090	**0.626**	−0.031	**0.334**	−0.097	0.097	**−0.497**	**−0.521**
R MSI		0.206	**0.511**	0.100	**0.325**	−0.147	−0.228	**−0.410**	**−0.357**
C MDA		0.060	**0.536**	**−0.463**	**0.349**	**−0.426**	0.219	**−0.458**	0.181
R MDA		0.178	**0.731**	−0.150	**0.330**	**−0.374**	−0.109	**−0.487**	−0.118
C Ca		−0.058	**−0.606**	0.099	−0.201	0.009	−0.204	**0.671**	0.290
R Ca		0.158	**−0.430**	0.198	**−0.349**	**0.357**	**−0.372**	**0.614**	−0.177
C Mg		0.150	**−0.585**	0.133	**−0.318**	−0.150	−0.066	**0.552**	0.284
R Mg		0.015	**−0.537**	0.222	−0.157	**0.567**	**−0.312**	**0.394**	−0.192
C Fe		−0.279	**−0.714**	**0.410**	−0.123	**0.375**	0.071	0.309	−0.049
R Fe		0.120	**−0.665**	0.107	**−0.405**	0.294	−0.118	**0.503**	0.163
C K		0.250	**−0.594**	0.291	−0.228	**0.396**	**−0.423**	0.310	−0.001
R K		0.097	**−0.572**	**0.361**	−0.119	0.004	**−0.417**	**0.446**	0.198
C Na		0.085	**0.731**	−0.262	0.280	**−0.440**	0.110	**−0.412**	−0.091
R Na		−0.023	**0.732**	−0.118	0.227	−0.280	0.037	−0.299	−0.275
C Na/K		−0.069	**0.780**	−0.278	0.232	**−0.398**	0.200	**−0.366**	−0.100
R Na/K		−0.099	**0.760**	−0.2622	0.149	−0.166	0.250	**−0.374**	−0.257
TG%		0.027	**−0.568**	0.2434	−0.243	0.135	−0.270	**0.459**	0.216

Variables were centered around their means and normalized with a standard deviation of 1, *n* = 5. Values in bold represent significant correlations at the 0.05 level.

## Data Availability

Authors comply with the Convention on Biological Diversity and the Convention on the Trade in Endangered Species of Wild Fauna and Flora.

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
