# Peer review of "The Efficiency of Different Priming Agents for Improving Germination and Early Seedling Growth of Local Tunisian Barley under Salinity Stress"

_plants, 2021, doi:10.3390/plants10112264_

Round 1
Reviewer 1 Report
Dear Editor
I carefully read the new version of the manuscript and the authors' responses to my earlier comments. Basically, the authors' answers satisfy me; the corresponding corrections were made to the text of the manuscript. The presented manuscript is significantly improved in comparison with the first version. I believe that the main advantage of this study is its thoroughness, good discussion. Despite the fact that a very small set of physiological research methods was used in the work, the authors managed to obtain rather reliable results that can arouse interest among a large group of researchers and practitioners. I consider it possible to recommend this article in the presented form for publication in the journal.
With kind regards
Vladimir Kuznetsov
Author Response
Dear Pr. Bobo Wang,
Thank you for all your comments and remarks that allow me improve my MS in comparison with the old version. I am grateful also for your important notes and reviewer’s detailed comments.Would you find please below detailed responses to the Reviewer's and Editor’s comments. We sincerely hope that we have answered all of your requests and that our work is now suitable for publication in ‘Plants’.
Best Regards.
Answer to Reviewer 1 comments:
Dear Professor Vladimir Kuznetsov,
First of all I want tell Thank you very much for supporting me and guiding me step by step to improve the quality of my MS. Your words mean a lot for me. I am happy that I did it and I enhanced my article following your comments and suggestions. Thank you also for your encouragement and patience.
Kind Regards.

Reviewer 2 Report
The authors may not have received my previous comments. So I'll send it again. Please check it carefully, especially those marked in red.
I would like to draw your attention that I asked you to describe in details the salt tolerance mechanisms of H, vulgare and H maritnum in the intruduction as well as the priming agents used previously. I found many papers dealing with these topics. I’ve also linked several articles to help you, but after searching on the net, the authors can also find many articles.
In addition, I found that the Fig2 A nd B are not clear (especially the figure marks). I suggest using the marker system logically. For example: the lines for controls (without priming agents) can be black lines, the priming with CaCl2 can be red line, with KCl can be green line , while with KNO3 can be blue line. Controls without salt treatment can be continuous lines, values of 100mM NaCl treated plants can be dashed lines, while 200mM NaCl cab be dotted lines. In this way, eveything will be clear. Of course you can find other systems, but they should be logic.

Author Response
Dear Pr. Bobo Wang,
Thank you for all your comments and remarks that allow me improve my MS in comparison with the old version. I am grateful also for your important notes and reviewer’s detailed comments.Would you find please below detailed responses to the Reviewer's and Editor’s comments. We sincerely hope that we have answered all of your requests and that our work is now suitable for publication in ‘Plants’.
Best Regards.
Answer to Reviewer 2 comments:
We thank Reviewer 1 for the review and detailed comments. The evaluation helped us to improve the quality of the manuscript. Please find the revised manuscript version with our detailed responses to the Reviewer's comments.
The presentation of the manuscript was corrected below as the Referee suggested (all the modifications and additions were marked with Track changes).
Many investigations performed on H. maritimum and the salt tolerance mechanisms of H. maritimum and H. vulgare were not mentioned in the introduction. Only the fact, that H. maritimum is a halophyte plant while H. vulgare is a glycophyte.
As suggested by the referee salt tolerance mechanisms of H. maritimum and H. vulgare were mentioned and developed in the introduction using “Track change”
4.5. I am not sure that 20mg DW (app. 200mg FW) was enough for correct determination of mineral elements. Yes, this amount is enough for the extraction, but the results will not be correct. At least 0.5 -1 g DW should be use to get correct results. E.g. Analytical Methods for Atomic Absorption Spectroscopy
We usually use this protocole in our Lab, the essential is that the amount of plant material should be more than the solvant of extraction to be sure that we extraced the maximum of ions. Moreover the concentration of ions in both barley species was high. And may be for germination stage 1g DW for each replicates ( 5 repetitions) is a lot and it was hard to collect it especially when we applicated the stress, because one of the species is very small.
Please add the mg plant material used and the number of repetition for each measure parameters within all adeqaute subsection of MM section (not in statistics -4.8).
As suggested by referee it was done in MM section.
The remark was not this. As you collected the seeds of H. maritimum, you can not garante the homogenity of your plant materials.
After collection of plant materials from the same territory, we choose healthy and uniform size seeds to be sure from the homogeneity of seeds. Moreover our aim is to study the effect of priming under salt stress only at the physiologically level.
I can accept that the authors want to merge the Fig.1 A and B for statistical analysis, but in this case, but it should mention in figure legends. However, the results of statistical analysis does not seem correct. For instance, 100mMNaCl (at CaCl2 of H. vulgare ) is cd, but 200mM NaCl (at CaCl2 of H. maritimum) is de, but according to the STD they are not seem logic. ETC. Please check the data again. It would be useful to use error lines up and down (not only up).
For figure 1 Data were checked and statistical analysis were re done for each species alone not merged together. And as suggested by the referee we use error lines up and down (not only up) for all figures.
Such kind of explanation (did by authors) is speculation, because the authors did not measured the mentioned parameters such as NO content, antioxidant activities, selectivity of Ca chanlels, and so on.
These information can be used in a way that: When XV investigated … they, found that pretreatment with CaCl2 resulted in… . … It is possible that in our cases, similar processes occured, however, further investigations are necessery o prove it.
This suggestion was applicated for some information in my MS as the reviewer commented.
In addition,
It is still not clear why the CaCl2 give better protection against salt stress than KCl or KNO3?
Why the priming agents seemed more efficient at 100mM NaCl tha et 200mM in H. vulgare.
In discusion section I developped why CaCl2 give better protection and why seed priming is effecient at low salt concentration using Track change
How do the priming agents affect the Fe2+ , Mg2+ content?
The stimulation of nutrient content by seed priming could be explained by the induction of plasma membrane H+ ATPase activity which plays an important role in the transport of multiple ions such as Fe2+ , Mg2+
In addition, I found that the Fig2 A nd B are not clear (especially the figure marks). I suggest using the marker system logically. For example: the lines for controls (without priming agents) can be black lines, the priming with CaCl2 can be red line, with KCl can be green line , while with KNO3 can be blue line. Controls without salt treatment can be continuous lines, values of 100mM NaCl treated plants can be dashed lines, while 200mM NaCl cab be dotted lines. In this way, eveything will be clear. Of course you can find other systems, but they should be logic.
The quality of the figure was improved as suggested by the referee

This manuscript is a resubmission of an earlier submission. The following is a list of the peer review reports and author responses from that submission.
Round 1
Reviewer 1 Report
The article of Rim Ben Youssef and coauthors is devoted to the study of impact of some priming agents like KNO3, KCl and CaCl2 on seed germination and early seedling growth under salinity. The data obtained in this paper are well presented and have practical importance.
Some notes and questions:
1) Why exactly these concentrations of priming components (5 mM CaCl2, 2% KNO3 and 2% KCl) were used. Why were the different times of 20 and 40 hours used to prime the seeds in the case of CaCl2 and KNO3/KCl?
2) 99-102 lines “The effect of seed priming was more prominent on seeds of 99 both cultivars primed with CaCl2 and germinated with100 mM NaCl. It was in order of 100 22.4% and 16% for Hordeum vulgare and Hordeum maritimum, respectively, compared to 101 their controls.” According to Fig. 1 the effect of CaCl2 priming was more prominent in the case of 200 mM NaCl.
3) Fig.1,4,5,6,7,8 What do the letters (a,ab,ah,ef,c,b and another) above the posts mean?
4) Fig. 2,3 Statistics should be added.
5) Fig. 2 “P CaCl2/100mM NaCl” needs to be replaced with “CaCl2/100mM NaCl”.
6) Fig. 3 “KNO3+100mM NaCl” needs to be replaced with “KNO3+100mM NaCl”.
7) How do the authors explain the difference in the effects for the two barley crops? Нow do these species differ from each other?
8) Fig. 4,6,7,8,9,10 Legends should be below the diagram names.
9) Fig. 5 What is about Hordeum maritimum?
10) Fig. 6,7 What is the reason for the 10-fold difference between the dry weight of two crops?
11)
59 line “….tomato [13] and maize [14,15] .Many…” needs to be replaced with ““….tomato [13] and maize [14,15]. Many…”.
64 line “…sol[17,18,19] . Similar” needs to be replaced with “…sol [17,18,19]. Similar”
69 line “Additionally, kaya et al. revealed..” needs to be replaced with “Additionally, Kaya et al. revealed…”.
75 line “power of k+ ion” needs to be replaced with “power of K+ ion”.
564-565 lines “primed with 5mM of CaCl2 solution for 20 hours and 564 soaked with 2% of KNO3” needs to be replaced with “primed with 5 mM of CaCl2 solution for 20 hours or 564 soaked with 2% of KNO3”.
12) 579 line “This allows clearing:” It looks strange.
13) 609-610 lines Сheck ion designation. Do this throughout the text.
14) 619-621 Please, design what do the EC1 and EC2 mean.
15) 631 and 636 lines Please, check for spaces. Do this throughout the text.
Author Response
Dear Pr. Bobo Wang,
Thank you for having considered our manuscript (plants-1269312). Would you find below the revised version with detailed responses to the Reviewer's comments. We sincerely hope that we have answered all of your requests and that our work is now suitable for publication in «Plants".
Best Regards.
Corresponding author: Rim Ben Youssef; E-Mail: rim.benyoussef@hotmail.com
Laboratory of Extrêmophile Plants, Centre of Biotechnology of Borj-Cedria,
- 901, Hammam-Lif 2050, Tunisia.
Answer to Reviewer 1 comments:
We thank Reviewer 1 for the review and detailed comments. The evaluation helped us to improve the quality of the manuscript. Please find the revised manuscript version with our detailed responses to the Reviewer's comments.
The presentation of the manuscript was corrected below as the Referee suggested (all the modifications and additions were marked with Track changes).
Detailed Response to Reviewer 1 comments
- Why exactly these concentrations of priming components (5 mM CaCl2, 2% KNO3 and 2% KCl) were used. Why were the different times of 20 and 40 hours used to prime the seeds in the case of CaCl2 and KNO3/KCl?
Our answer:
A preliminary test was carried out, in fact, for both barley species, for each pretreatment agent several concentrations and several times were tested and then according to statistics analyses the time and concentration that gave a better stimulation of the germinal parameters and the growth traits of seeds were chosen for each agent. Results of the preliminary test showed that the effect of KNO3 was more pronounced with 2% at 40h; this of KCl was more prominent with 2% at 40h, however CaCl2 was more efficient with 5mM at 20h. Moreover, many previous studies testing priming with those agents, most of them worked with concentration in % for KCl and KNO3 and in mM for CaCl2. Concentrations and times of priming agents were chosen based on our own preliminary test later confirmed by comparison with previous investigations such as in wheat (Paul and Choudhury 1991), in sorghum (Shehzad, and al. 2012) and rice (Ruttanaruangboworn, et al. 2017).
99-102 lines “The effect of seed priming was more prominent on seeds of 99 both cultivars primed with CaCl2 and germinated with 100 mM NaCl. It was in order of 100 22.4% and 16% for Hordeum vulgare and Hordeum maritimum, respectively, compared to 101 their controls.” According to Fig.1 the effect of CaCl2 priming was more prominent in the case of 200 mM NaCl.
Our answer:
As the Referee recommended, we checked the results presented by figure 1 again and we corrected this information as follow. So,
“This beneficial effect was agent and salt concentration dependent. The effect of seed priming was more prominent on seeds of both cultivars primed with CaCl2 and germinated with100 mM NaCl. It was in order of 22.4% and 16% for Hordeum vulgare and Hordeum maritimum, respectively, compared to their controls.” Was replaced by “Among the priming agent, CaCl2 seemed to be the most efficient protector causing an increase by 26 and 79% in FG%, respectively in both species. While, without priming agents the 200 mM NaCl caused 47% and 24.5% reduction of FG% for H. vulgare and H. maritimum, respectively.”
- 1,4,5,6,7,8 What do the letters (a,ab,ah,ef,c,b and another) above the posts mean?
Our answer:
These letters belong to the statistical analysis. Our objective is to compare the effect of priming agents on both barley species exposed to salt concentrations. The effect is significant when the letters are different.
- 2,3 Statistics should be added.
Our answer:
Figure 2 was deleted as proposed by your colleague and statistics were added in Figure 3 which belongs to the MDG turns in this version to figure 2.
- 2 “P CaCl2/100mM NaCl” needs to be replaced with “CaCl2/100mM NaCl”.
Our answer:
Figure 2 was deleted as proposed Reviewer 3.
- 3 “KNO3+100mM NaCl” needs to be replaced with “KNO3+100mM NaCl”.
Our answer:
As the Referee recommended, it was replaced on the figure 3 which turns in this version to figure 2.
- How do the authors explain the difference in the effects for the two barley crops? Нow do these species differ from each other?
Our answer:
The two barley species differ from each other by their origins. In fact Hordeum maritimum is a facultative halophyte, a wild cereal However Hordeum vulgare is a cultivated glycophyte. The first species is able to tolerate high salt concentration but the second one is relatively tolerant.
- 4,6,7,8,9,10 Legends should be below the diagram names.
Our answer:
As the Referee recommended, it was done for all figures. Just the number of figures changed because I deleted some of them as proposed by the Referee 3.
- 5 What is about Hordeum maritimum?
Our answer:
We presented root number just for Hordeum vulgare because Hordeum maritimum have just a principal root it is not enough to count it and analyze this parameter. Since our objective is to compare both species we preferred to delete this parameter according to your comment.
- 6, 7 What is the reason for the 10-fold difference between the dry weight of two crops?
Our answer:
The difference between the dry weights of two crops is due the fact that Hordeum maritimum tissues are smaller than Hordeum vulgare. I am comparing two species one of them is a glycophyte cultivated (Hordeum vulgare) while the other one is an halophyte wild or spontaneous (Hordeum maritimum). So, as you suggested that the following paragraph was added (see introduction)
11)
59 line “….tomato [13] and maize [14,15] .Many…” needs to be replaced with ““….tomato [13] and maize [14,15]. Many…”.
Our answer:
As the Referee noted, tomato [13] and maize [14,15] .Many…” was replaced with ““….tomato [13] and maize [14,15]. Many…”.
64 line “…sol[17,18,19] . Similar” needs to be replaced with “…sol [17,18,19]. Similar”
Our answer:
As the Referee noted, “…sol[17,18,19] . Similar” was replaced with “…sol [17, 18, 19]. Similar”
69 line “Additionally, kaya et al. revealed..” needs to be replaced with “Additionally, Kaya et al. revealed…”.
Our answer:
As the Referee noted, “Additionally, kaya et al. revealed..” was replaced with “Additionally, Kaya et al. revealed…”.
75 line “power of k+ ion” needs to be replaced with “power of K+ ion”.
Our answer:
As the Referee noted, “power of k+ ion” was replaced with“ power of K+ ion”.
564-565 lines “primed with 5mM of CaCl2 solution for 20 hours and 564 soaked with 2% of KNO3” needs to be replaced with “primed with 5 mM of CaCl2 solution for 20 hours or 564 soaked with 2% of KNO3”.
Our answer:
As the Referee noted, “primed with 5mM of CaCl2 solution for 20 hours and 564 soaked with 2% of KNO3” was replaced with “primed with 5 mM of CaCl2 solution for 20 hours or 564 soaked with 2% of KNO3”.
- 579 line “This allows clearing:” It looks strange.
Our answer:
“This allows clearing:” It was deleted as the referee noted.
- 609-610 lines Сheck ion designation. Do this throughout the text.
Our answer:
As the Referee noted, ion designation was checked.
- 619-621 Please, design what do the EC1 and EC2 mean.
Our answer:
As the Referee noted, it was done.
- 631 and 636 lines Please, check for spaces. Do this throughout the text.
Our answer:
As the Referee noted, it was done.
Answer to Reviewer 2 comments:
We thank the Reviewer for their careful reading of the manuscript and their constructive remarks. We sincerely appreciate all the valuable comments and suggestions which helped us to improve the quality of our manuscript.
Detailed Response to Reviewer 2 comments
- I am questioned by the data according to which the exposure of seeds to sodium chloride solutions is accompanied by an increase in their water content. The authors explain this by the fact that the seeds absorb salt, which leads to a drop in the water potential in the seed cells and the absorption of water. Everything that I know in this field of science, the exposure of plant objects to salt solution is accompanied by dehydration. It seems to me that the proposed explanation of the data obtained is not sufficient.
Our answer:
Our results showed that salt stress increased water content on Hordeum vulgare seeds in both tissues (Fig5). These results are consistent with those reported by Debez et al. (2004) in Cakile maritima, M’sehli et al (2011) in Medicago ciliaris and Boukari et al. (2019) in Medicago sativa who noted that WC increased under salt stress in Gabes ecotype shoots as well as roots. This phenomenon could be due the osmotic potential of cells which was probably lowered so that the water supply was not disrupted by external salinity.
It is not clear why the authors used filter paper when incubating seeds in Petri dishes. The filter paper is not so indifferent. Many years ago prof. X Borris of the University of Greifswald has shown that there is a factor in the filter paper (called the "paper factor") that affects seed germination.
Our answer:
One of our objectives is to analyze ion content, so we used a specific paper called ashless Filter paper. This type of paper doesn’t interfere the germination process and doesn’t contain element that could be absorbed by seeds.
It is necessary to unify the designation of the concentration of solutions. The work uses three salts; the concentration of one of them is expressed in moles, and the other two salts - in percent.
Our answer:
A preliminary test was carried out, in fact, for both barley species, for each pretreatment agent several concentrations and several times were tested and then according to statistics analyses the time and concentration that gave a better stimulation of the germinal parameters and the growth traits of seeds were chosen for each agent. Results of the preliminary test showed that the effect of KNO3 was more pronounced with 2% at 40h; this of KCl was more prominent with 2% at 40h, however CaCl2 was more efficient with 5mM at 20h.
Answer to Reviewer 3 comments:
We thank the Reviewer for their careful reading of the manuscript and their constructive remarks. We sincerely appreciate all the valuable comments and suggestions which helped us to improve the quality of our manuscript.
Detailed Response to Reviewer 3 comments
I think, the salt stress responses of both H. vulgare (with different accessions) and H marinum (H. maritimum) have been widely investigated even during germination, although the authors did not mention these results in the introduction. H. maritimum, a wild seaside barley, is a halophyte plant while H. vulgare is glycophyte and their salt tolerance mechanism are completely different. This information should be mentioned in the introduction.
Our answer:
As referee noted, the requested informations was added in the introduction.
-It seems that the effect of priming agents were efficient in H. vulgare (DW), while were unaffected in H. maritimum probably due to the different salt tolerance mechanisms. It should discuss better these differences.
Our answer:
As referee noted: This information was explained in the discussion part (See line )
L85-86 and L557: H vulgare and H maritimum are two species, not two cultivars. It should be given cultivar name of H. vulgare used in the experiments (as see in https://www.publish.csiro.au/fp/FP18046) . In addition, and as it was mentioned in the MM section, the seeds of H maritimum was collected from …. Are you sure that these seeds originated from genetically homogen population?
Our answer:
Our objective is to investigate the effect of priming with different agents on the germination of seeds of both barley species under salinity stress. We are focused only on the physiological responses so we collected seeds from the same territories. We are not working in genetic traits.
Although the authors presented a large number of data, these data are originated from different combinations of a germination test. Thus, it is evident that all data lead to the same conclusion. Otherwise, I do not think, that all data are necessary to present. If you think, that each figure provides new information, please discuss them in details, otherwise you can eliminate some of them (e.g. Fig.2, Fig.6,).
Our answer:
AS referee proposed, Fig 2 and Fig 6 were eliminated.
Results: The results section contains many unnecessary (long sentences with little information). Such as between L92-102; L103-107, 108, 124-126, 131, 157-161…195-207, 210-214 (What a surprise! – it is well known information), 332-336, 442-445 and so on. These should be eliminated or shortened. This type of repetitions makes the text boring and the readers lose their interest since they do not get real new information. Less is often more!
For example: L92-102: Results showed that salinity affected negatively the FG % of unprimed seeds on both barley cultivars. This effect depended on the salt concentration and the cultivar Indeed, side impacts were more pronounced when seeds were germinated with 200 mM NaCl…..:
You can write: Priming with KCl, KNO3 or CaCl2 did not modify the germination rate without salt stress, but alleviated the effect of salt stress caused by either 100 or 200 mM NaCl both in H maritimum and H. vulgare. Among the priming agent, CaCl2 seemed the most efficient protector causing only 22.4 and 16% reduction in FG%, while without priming agents the 200mM NaCl caused 47% and 24.5% reduction of FG% for H. vulgare and H. maritimum, respectively.
Instead of L157-176, you can write:
The coleoptiles and radicles length of unprimed seeds in Hordeum vulgare decreased by 65.4% and 66.7% when the seeds were exposed to 100 or 200mM salt stress, while those of Hordeum maritimum were reduced by 27% and 30.2%, respectively, compared to their controls (Fig. 4). The priming agents, especiallly CaCl2 enhanced growth of radicle in H. maritime and coleoptyle length in both species with or without salt treatement. …..
Our answer:
Results of all parameters were shortened as recommended by referee. See Results section.
If you write “This effect depended on the salt concentration and the cultivar” or “ this beneficial effect was agent and salt concentration dependent” you should prove them statistically for example by factorial analysis of variance (ANOVA)
Our answer:
This sentence of dependence was eliminated.
The statistics of Fig. 1 seems strange. On Fig.1A where is g , if there is “h” . On Fig.1B , there is no “a” , only ab. Why it is separted from “b”. Where is “A”? Etc. Please check the statistics again. I think, it is not correct to make the statistics together for both species, but present the values on separate figures (Fig.1A and B). Alternatively, write it in the Figure legends to be evident.
Our answer:
Since the objective of our study is to compare the response of primed and unprimed seeds to salinity, the statistical analysis was done together. In one hand, we bring them together to facilitate the comparison. In the other hand, to more clarify, we were obliged to divide the results in both figure a and b.
Fig.2 presents the germination % vs time (days) not LT and T50 as indicated in L104-106. In addition, why are not presented the values of KCl, KNO3 and CaCl2/0 mM NaCl? on Fig.2 and Fig.3 too.
Our answer:
As recommended by referee, values of KCl, KNO3 and CaCl2/0 mM NaCl are now presented in fig.3. Fig.2 was eliminated as the referee recommended.
Fig.5: Please use “Radicle length (cm)” and Coleoptile length (cm) in the adequate axis (A and C) of Fig.5.
Our answer:
As referee noted, it was done.
Fig.8 the data of WC seems strange. In general the WC decrease when the plants are exposed to NaCl especially in glycophyte plants. In 4.4 it is not mentioned how many plants were used for determination of WC. Perhaps the amount of plant materials was too low. I suggest repeating this type of measurement. It is quick. Alternatively, explain better how it is possible that the WC increase after NaCl in H. vulgare, but not in H. maritimum (which is probably a salt tolerant genotype). 481-485 seem seem to contradict the former lines (474-476 –it is normal phenomenon under salt stress).
Our answer:
We checked again our results, they are okay. In fact H. Vulgare is the best cultivated cereal able to tolerate salt and drought. These results are consistent with those reported by Debez et al. (2004) in Cakile maritima, M’sehli et al (2011) in Medicago ciliaris and Boukari et al. (2019) in Medicago sativa who noted that WC increased under salt stress in Gabes ecotype shoots as well as roots. This phenomenon could be due the osmotic potential of cells which was probably lowered so that the water supply was not disrupted by external salinity.
In addition, please label the Y axis more precisely (e.g. WC of radicle) . It is true for Fig.9 too.
Our answer:
Done, as referee noted.
Discussion
More explanation were added
MM section:
Please add the name of cultivar of H. vulgare used in the experiments.
Our answer:
Done
Please explain better why these concentrations (5mM CaCl2 and 2% KCl and 2% KNO3) and time (20 h or 40h) were used as priming. Why these conditions were used?
Our answer:
A preliminary test was carried out, in fact, for both barley species, for each pretreatment agent several concentrations and several times were tested and then according to statistics analyses the time and concentration that gave a better stimulation of the germinal parameters and the growth traits of seeds were chosen for each agent. Results of the preliminary test showed that the effect of KNO3 was more pronounced with 2% at 40h; this of KCl was more prominent with 2% at 40h, however CaCl2 was more efficient with 5mM at 20h. Moreover, many previous studies testing priming with those agents, most of them worked with concentration in % for KCl and KNO3and in mM for CaCl2.
4.5. I am not sure that 20mg DW (app. 200mg FW) was enough for correct determination of mineral elements.
We used 20 mg of DW and the volume of liquid of extraction is less so we were able to extract the maximum of ions. The same protocol is always used in many investigations.
4.6. Membrane integrity was estimated by measuring the MSI using a digital conductivity meter (type Metrohom). Initially, we cut 0.2 g of fresh material and placed it in a falcon with distilled water (10 ml), after then, we heated it at 32 °C for 30 minutes and finally, we recorded the electrical conductivity (EC1). The conductivity was also assayed after placing the samples at 100 °C for 2 hours (EC2). The MSI was measured according to Dionioese and Tobita using the following formula [29]: Please add the number of repetition for each measure parameters within the same subsection (not in statistics -4.8). It is difficult to follow. In addition, please add how many Petri dishes were used for each treatment.
Our answer:
Done
I think, only a germination test is insufficient for publication in Plants. I suggest following this type of experiment with older plants to demonstrate that the priming could be efficient at later developmental stages too and study what kind of mechanisms operate during the priming in H. vulgare or H. maritimum. (Although I believe that you can find information about it in many papers.
Our answer:
Our objective is to study seed physiological and biochemical response at germinative and early seedling growth and especially to evaluate the possible remedial effect of priming with different agents under salinity conditions. As you suggested, this work will be continued by an investigation at the vegetative stage.
Dear Pr. Bobo Wang,
Thank you for having considered our manuscript (plants-1269312). Would you find below the revised version with detailed responses to the Reviewer's comments. We sincerely hope that we have answered all of your requests and that our work is now suitable for publication in «Plants".
Best Regards.
Corresponding author: Rim Ben Youssef; E-Mail: rim.benyoussef@hotmail.com
Laboratory of Extrêmophile Plants, Centre of Biotechnology of Borj-Cedria,
- 901, Hammam-Lif 2050, Tunisia.
Answer to Reviewer 1 comments:
We thank Reviewer 1 for the review and detailed comments. The evaluation helped us to improve the quality of the manuscript. Please find the revised manuscript version with our detailed responses to the Reviewer's comments.
The presentation of the manuscript was corrected below as the Referee suggested (all the modifications and additions were marked with Track changes).
Detailed Response to Reviewer 1 comments
- Why exactly these concentrations of priming components (5 mM CaCl2, 2% KNO3 and 2% KCl) were used. Why were the different times of 20 and 40 hours used to prime the seeds in the case of CaCl2 and KNO3/KCl?
Our answer:
A preliminary test was carried out, in fact, for both barley species, for each pretreatment agent several concentrations and several times were tested and then according to statistics analyses the time and concentration that gave a better stimulation of the germinal parameters and the growth traits of seeds were chosen for each agent. Results of the preliminary test showed that the effect of KNO3 was more pronounced with 2% at 40h; this of KCl was more prominent with 2% at 40h, however CaCl2 was more efficient with 5mM at 20h. Moreover, many previous studies testing priming with those agents, most of them worked with concentration in % for KCl and KNO3 and in mM for CaCl2. Concentrations and times of priming agents were chosen based on our own preliminary test later confirmed by comparison with previous investigations such as in wheat (Paul and Choudhury 1991), in sorghum (Shehzad, and al. 2012) and rice (Ruttanaruangboworn, et al. 2017).
99-102 lines “The effect of seed priming was more prominent on seeds of 99 both cultivars primed with CaCl2 and germinated with 100 mM NaCl. It was in order of 100 22.4% and 16% for Hordeum vulgare and Hordeum maritimum, respectively, compared to 101 their controls.” According to Fig.1 the effect of CaCl2 priming was more prominent in the case of 200 mM NaCl.
Our answer:
As the Referee recommended, we checked the results presented by figure 1 again and we corrected this information as follow. So,
“This beneficial effect was agent and salt concentration dependent. The effect of seed priming was more prominent on seeds of both cultivars primed with CaCl2 and germinated with100 mM NaCl. It was in order of 22.4% and 16% for Hordeum vulgare and Hordeum maritimum, respectively, compared to their controls.” Was replaced by “Among the priming agent, CaCl2 seemed to be the most efficient protector causing an increase by 26 and 79% in FG%, respectively in both species. While, without priming agents the 200 mM NaCl caused 47% and 24.5% reduction of FG% for H. vulgare and H. maritimum, respectively.”
- 1,4,5,6,7,8 What do the letters (a,ab,ah,ef,c,b and another) above the posts mean?
Our answer:
These letters belong to the statistical analysis. Our objective is to compare the effect of priming agents on both barley species exposed to salt concentrations. The effect is significant when the letters are different.
- 2,3 Statistics should be added.
Our answer:
Figure 2 was deleted as proposed by your colleague and statistics were added in Figure 3 which belongs to the MDG turns in this version to figure 2.
- 2 “P CaCl2/100mM NaCl” needs to be replaced with “CaCl2/100mM NaCl”.
Our answer:
Figure 2 was deleted as proposed Reviewer 3.
- 3 “KNO3+100mM NaCl” needs to be replaced with “KNO3+100mM NaCl”.
Our answer:
As the Referee recommended, it was replaced on the figure 3 which turns in this version to figure 2.
- How do the authors explain the difference in the effects for the two barley crops? Нow do these species differ from each other?
Our answer:
The two barley species differ from each other by their origins. In fact Hordeum maritimum is a facultative halophyte, a wild cereal However Hordeum vulgare is a cultivated glycophyte. The first species is able to tolerate high salt concentration but the second one is relatively tolerant.
- 4,6,7,8,9,10 Legends should be below the diagram names.
Our answer:
As the Referee recommended, it was done for all figures. Just the number of figures changed because I deleted some of them as proposed by the Referee 3.
- 5 What is about Hordeum maritimum?
Our answer:
We presented root number just for Hordeum vulgare because Hordeum maritimum have just a principal root it is not enough to count it and analyze this parameter. Since our objective is to compare both species we preferred to delete this parameter according to your comment.
- 6, 7 What is the reason for the 10-fold difference between the dry weight of two crops?
Our answer:
The difference between the dry weights of two crops is due the fact that Hordeum maritimum tissues are smaller than Hordeum vulgare. I am comparing two species one of them is a glycophyte cultivated (Hordeum vulgare) while the other one is an halophyte wild or spontaneous (Hordeum maritimum). So, as you suggested that the following paragraph was added (see introduction)
11)
59 line “….tomato [13] and maize [14,15] .Many…” needs to be replaced with ““….tomato [13] and maize [14,15]. Many…”.
Our answer:
As the Referee noted, tomato [13] and maize [14,15] .Many…” was replaced with ““….tomato [13] and maize [14,15]. Many…”.
64 line “…sol[17,18,19] . Similar” needs to be replaced with “…sol [17,18,19]. Similar”
Our answer:
As the Referee noted, “…sol[17,18,19] . Similar” was replaced with “…sol [17, 18, 19]. Similar”
69 line “Additionally, kaya et al. revealed..” needs to be replaced with “Additionally, Kaya et al. revealed…”.
Our answer:
As the Referee noted, “Additionally, kaya et al. revealed..” was replaced with “Additionally, Kaya et al. revealed…”.
75 line “power of k+ ion” needs to be replaced with “power of K+ ion”.
Our answer:
As the Referee noted, “power of k+ ion” was replaced with“ power of K+ ion”.
564-565 lines “primed with 5mM of CaCl2 solution for 20 hours and 564 soaked with 2% of KNO3” needs to be replaced with “primed with 5 mM of CaCl2 solution for 20 hours or 564 soaked with 2% of KNO3”.
Our answer:
As the Referee noted, “primed with 5mM of CaCl2 solution for 20 hours and 564 soaked with 2% of KNO3” was replaced with “primed with 5 mM of CaCl2 solution for 20 hours or 564 soaked with 2% of KNO3”.
- 579 line “This allows clearing:” It looks strange.
Our answer:
“This allows clearing:” It was deleted as the referee noted.
- 609-610 lines Сheck ion designation. Do this throughout the text.
Our answer:
As the Referee noted, ion designation was checked.
- 619-621 Please, design what do the EC1 and EC2 mean.
Our answer:
As the Referee noted, it was done.
- 631 and 636 lines Please, check for spaces. Do this throughout the text.
Our answer:
As the Referee noted, it was done.
Answer to Reviewer 2 comments:
We thank the Reviewer for their careful reading of the manuscript and their constructive remarks. We sincerely appreciate all the valuable comments and suggestions which helped us to improve the quality of our manuscript.
Detailed Response to Reviewer 2 comments
- I am questioned by the data according to which the exposure of seeds to sodium chloride solutions is accompanied by an increase in their water content. The authors explain this by the fact that the seeds absorb salt, which leads to a drop in the water potential in the seed cells and the absorption of water. Everything that I know in this field of science, the exposure of plant objects to salt solution is accompanied by dehydration. It seems to me that the proposed explanation of the data obtained is not sufficient.
Our answer:
Our results showed that salt stress increased water content on Hordeum vulgare seeds in both tissues (Fig5). These results are consistent with those reported by Debez et al. (2004) in Cakile maritima, M’sehli et al (2011) in Medicago ciliaris and Boukari et al. (2019) in Medicago sativa who noted that WC increased under salt stress in Gabes ecotype shoots as well as roots. This phenomenon could be due the osmotic potential of cells which was probably lowered so that the water supply was not disrupted by external salinity.
It is not clear why the authors used filter paper when incubating seeds in Petri dishes. The filter paper is not so indifferent. Many years ago prof. X Borris of the University of Greifswald has shown that there is a factor in the filter paper (called the "paper factor") that affects seed germination.
Our answer:
One of our objectives is to analyze ion content, so we used a specific paper called ashless Filter paper. This type of paper doesn’t interfere the germination process and doesn’t contain element that could be absorbed by seeds.
It is necessary to unify the designation of the concentration of solutions. The work uses three salts; the concentration of one of them is expressed in moles, and the other two salts - in percent.
Our answer:
A preliminary test was carried out, in fact, for both barley species, for each pretreatment agent several concentrations and several times were tested and then according to statistics analyses the time and concentration that gave a better stimulation of the germinal parameters and the growth traits of seeds were chosen for each agent. Results of the preliminary test showed that the effect of KNO3 was more pronounced with 2% at 40h; this of KCl was more prominent with 2% at 40h, however CaCl2 was more efficient with 5mM at 20h.
Answer to Reviewer 3 comments:
We thank the Reviewer for their careful reading of the manuscript and their constructive remarks. We sincerely appreciate all the valuable comments and suggestions which helped us to improve the quality of our manuscript.
Detailed Response to Reviewer 3 comments
I think, the salt stress responses of both H. vulgare (with different accessions) and H marinum (H. maritimum) have been widely investigated even during germination, although the authors did not mention these results in the introduction. H. maritimum, a wild seaside barley, is a halophyte plant while H. vulgare is glycophyte and their salt tolerance mechanism are completely different. This information should be mentioned in the introduction.
Our answer:
As referee noted, the requested informations was added in the introduction.
-It seems that the effect of priming agents were efficient in H. vulgare (DW), while were unaffected in H. maritimum probably due to the different salt tolerance mechanisms. It should discuss better these differences.
Our answer:
As referee noted: This information was explained in the discussion part (See line )
L85-86 and L557: H vulgare and H maritimum are two species, not two cultivars. It should be given cultivar name of H. vulgare used in the experiments (as see in https://www.publish.csiro.au/fp/FP18046) . In addition, and as it was mentioned in the MM section, the seeds of H maritimum was collected from …. Are you sure that these seeds originated from genetically homogen population?
Our answer:
Our objective is to investigate the effect of priming with different agents on the germination of seeds of both barley species under salinity stress. We are focused only on the physiological responses so we collected seeds from the same territories. We are not working in genetic traits.
Although the authors presented a large number of data, these data are originated from different combinations of a germination test. Thus, it is evident that all data lead to the same conclusion. Otherwise, I do not think, that all data are necessary to present. If you think, that each figure provides new information, please discuss them in details, otherwise you can eliminate some of them (e.g. Fig.2, Fig.6,).
Our answer:
AS referee proposed, Fig 2 and Fig 6 were eliminated.
Results: The results section contains many unnecessary (long sentences with little information). Such as between L92-102; L103-107, 108, 124-126, 131, 157-161…195-207, 210-214 (What a surprise! – it is well known information), 332-336, 442-445 and so on. These should be eliminated or shortened. This type of repetitions makes the text boring and the readers lose their interest since they do not get real new information. Less is often more!
For example: L92-102: Results showed that salinity affected negatively the FG % of unprimed seeds on both barley cultivars. This effect depended on the salt concentration and the cultivar Indeed, side impacts were more pronounced when seeds were germinated with 200 mM NaCl…..:
You can write: Priming with KCl, KNO3 or CaCl2 did not modify the germination rate without salt stress, but alleviated the effect of salt stress caused by either 100 or 200 mM NaCl both in H maritimum and H. vulgare. Among the priming agent, CaCl2 seemed the most efficient protector causing only 22.4 and 16% reduction in FG%, while without priming agents the 200mM NaCl caused 47% and 24.5% reduction of FG% for H. vulgare and H. maritimum, respectively.
Instead of L157-176, you can write:
The coleoptiles and radicles length of unprimed seeds in Hordeum vulgare decreased by 65.4% and 66.7% when the seeds were exposed to 100 or 200mM salt stress, while those of Hordeum maritimum were reduced by 27% and 30.2%, respectively, compared to their controls (Fig. 4). The priming agents, especiallly CaCl2 enhanced growth of radicle in H. maritime and coleoptyle length in both species with or without salt treatement. …..
Our answer:
Results of all parameters were shortened as recommended by referee. See Results section.
If you write “This effect depended on the salt concentration and the cultivar” or “ this beneficial effect was agent and salt concentration dependent” you should prove them statistically for example by factorial analysis of variance (ANOVA)
Our answer:
This sentence of dependence was eliminated.
The statistics of Fig. 1 seems strange. On Fig.1A where is g , if there is “h” . On Fig.1B , there is no “a” , only ab. Why it is separted from “b”. Where is “A”? Etc. Please check the statistics again. I think, it is not correct to make the statistics together for both species, but present the values on separate figures (Fig.1A and B). Alternatively, write it in the Figure legends to be evident.
Our answer:
Since the objective of our study is to compare the response of primed and unprimed seeds to salinity, the statistical analysis was done together. In one hand, we bring them together to facilitate the comparison. In the other hand, to more clarify, we were obliged to divide the results in both figure a and b.
Fig.2 presents the germination % vs time (days) not LT and T50 as indicated in L104-106. In addition, why are not presented the values of KCl, KNO3 and CaCl2/0 mM NaCl? on Fig.2 and Fig.3 too.
Our answer:
As recommended by referee, values of KCl, KNO3 and CaCl2/0 mM NaCl are now presented in fig.3. Fig.2 was eliminated as the referee recommended.
Fig.5: Please use “Radicle length (cm)” and Coleoptile length (cm) in the adequate axis (A and C) of Fig.5.
Our answer:
As referee noted, it was done.
Fig.8 the data of WC seems strange. In general the WC decrease when the plants are exposed to NaCl especially in glycophyte plants. In 4.4 it is not mentioned how many plants were used for determination of WC. Perhaps the amount of plant materials was too low. I suggest repeating this type of measurement. It is quick. Alternatively, explain better how it is possible that the WC increase after NaCl in H. vulgare, but not in H. maritimum (which is probably a salt tolerant genotype). 481-485 seem seem to contradict the former lines (474-476 –it is normal phenomenon under salt stress).
Our answer:
We checked again our results, they are okay. In fact H. Vulgare is the best cultivated cereal able to tolerate salt and drought. These results are consistent with those reported by Debez et al. (2004) in Cakile maritima, M’sehli et al (2011) in Medicago ciliaris and Boukari et al. (2019) in Medicago sativa who noted that WC increased under salt stress in Gabes ecotype shoots as well as roots. This phenomenon could be due the osmotic potential of cells which was probably lowered so that the water supply was not disrupted by external salinity.
In addition, please label the Y axis more precisely (e.g. WC of radicle) . It is true for Fig.9 too.
Our answer:
Done, as referee noted.
Discussion
More explanation were added
MM section:
Please add the name of cultivar of H. vulgare used in the experiments.
Our answer:
Done
Please explain better why these concentrations (5mM CaCl2 and 2% KCl and 2% KNO3) and time (20 h or 40h) were used as priming. Why these conditions were used?
Our answer:
A preliminary test was carried out, in fact, for both barley species, for each pretreatment agent several concentrations and several times were tested and then according to statistics analyses the time and concentration that gave a better stimulation of the germinal parameters and the growth traits of seeds were chosen for each agent. Results of the preliminary test showed that the effect of KNO3 was more pronounced with 2% at 40h; this of KCl was more prominent with 2% at 40h, however CaCl2 was more efficient with 5mM at 20h. Moreover, many previous studies testing priming with those agents, most of them worked with concentration in % for KCl and KNO3and in mM for CaCl2.
4.5. I am not sure that 20mg DW (app. 200mg FW) was enough for correct determination of mineral elements.
We used 20 mg of DW and the volume of liquid of extraction is less so we were able to extract the maximum of ions. The same protocol is always used in many investigations.
4.6. Membrane integrity was estimated by measuring the MSI using a digital conductivity meter (type Metrohom). Initially, we cut 0.2 g of fresh material and placed it in a falcon with distilled water (10 ml), after then, we heated it at 32 °C for 30 minutes and finally, we recorded the electrical conductivity (EC1). The conductivity was also assayed after placing the samples at 100 °C for 2 hours (EC2). The MSI was measured according to Dionioese and Tobita using the following formula [29]: Please add the number of repetition for each measure parameters within the same subsection (not in statistics -4.8). It is difficult to follow. In addition, please add how many Petri dishes were used for each treatment.
Our answer:
Done
I think, only a germination test is insufficient for publication in Plants. I suggest following this type of experiment with older plants to demonstrate that the priming could be efficient at later developmental stages too and study what kind of mechanisms operate during the priming in H. vulgare or H. maritimum. (Although I believe that you can find information about it in many papers.
Our answer:
Our objective is to study seed physiological and biochemical response at germinative and early seedling growth and especially to evaluate the possible remedial effect of priming with different agents under salinity conditions. As you suggested, this work will be continued by an investigation at the vegetative stage.

Reviewer 2 Report
Dear Authors
I read this manuscript with interest. The work is relevant for understanding the physiological mechanisms of seed priming. The authors compared the protective effect of pretreatment of barley seeds with solutions of CaCl2, KCl and KNO3 and showed that calcium chloride exhibits the greatest protective effect. This is expected, since more than 100 years ago in Central Asia, peasants applied calcium salts to saline soils to reduce the toxic effect of NaCl. At the same time, such a detailed comparison of the effects of 3 compounds on the germination of the canopy and the growth of seedlings has not yet been performed. A feature of the work is that the authors did not use any modern research method and, nevertheless, the work looks quite solid. The authors obtained fairly reliable experimental data. I have no doubt that the publication of the manuscript will arouse some interest among readers. It shows that even with the simplest approaches, decent work can be done.
Remarks.
1. I am questioned by the data according to which the exposure of seeds to sodium chloride solutions is accompanied by an increase in their water content. The authors explain this by the fact that the seeds absorb salt, which leads to a drop in the water potential in the seed cells and the absorption of water. Everything that I know in this field of science, the exposure of plant objects to salt solution is accompanied by dehydration. It seems to me that the proposed explanation of the data obtained is not sufficient.
2. It is not clear why the authors used filter paper when incubating seeds in Petri dishes. The filter paper is not so indifferent. Many years ago prof. X Borris of the University of Greifswald has shown that there is a factor in the filter paper (called the "paper factor") that affects seed germination.
3. It is necessary to unify the designation of the concentration of solutions. The work uses three salts, the concentration of one of them is expressed in moles, and the other two salts - in percent.
I think that the article can be recommended for publication after a little revision.
Author Response
Dear Pr. Bobo Wang,
Thank you for having considered our manuscript (plants-1269312). Would you find below the revised version with detailed responses to the Reviewer's comments. We sincerely hope that we have answered all of your requests and that our work is now suitable for publication in «Plants".
Best Regards.
Corresponding author: Rim Ben Youssef; E-Mail: rim.benyoussef@hotmail.com
Laboratory of Extrêmophile Plants, Centre of Biotechnology of Borj-Cedria,
- 901, Hammam-Lif 2050, Tunisia.
Answer to Reviewer 1 comments:
We thank Reviewer 1 for the review and detailed comments. The evaluation helped us to improve the quality of the manuscript. Please find the revised manuscript version with our detailed responses to the Reviewer's comments.
The presentation of the manuscript was corrected below as the Referee suggested (all the modifications and additions were marked with Track changes).
Detailed Response to Reviewer 1 comments
- Why exactly these concentrations of priming components (5 mM CaCl2, 2% KNO3 and 2% KCl) were used. Why were the different times of 20 and 40 hours used to prime the seeds in the case of CaCl2 and KNO3/KCl?
Our answer:
A preliminary test was carried out, in fact, for both barley species, for each pretreatment agent several concentrations and several times were tested and then according to statistics analyses the time and concentration that gave a better stimulation of the germinal parameters and the growth traits of seeds were chosen for each agent. Results of the preliminary test showed that the effect of KNO3 was more pronounced with 2% at 40h; this of KCl was more prominent with 2% at 40h, however CaCl2 was more efficient with 5mM at 20h. Moreover, many previous studies testing priming with those agents, most of them worked with concentration in % for KCl and KNO3 and in mM for CaCl2. Concentrations and times of priming agents were chosen based on our own preliminary test later confirmed by comparison with previous investigations such as in wheat (Paul and Choudhury 1991), in sorghum (Shehzad, and al. 2012) and rice (Ruttanaruangboworn, et al. 2017).
99-102 lines “The effect of seed priming was more prominent on seeds of 99 both cultivars primed with CaCl2 and germinated with 100 mM NaCl. It was in order of 100 22.4% and 16% for Hordeum vulgare and Hordeum maritimum, respectively, compared to 101 their controls.” According to Fig.1 the effect of CaCl2 priming was more prominent in the case of 200 mM NaCl.
Our answer:
As the Referee recommended, we checked the results presented by figure 1 again and we corrected this information as follow. So,
“This beneficial effect was agent and salt concentration dependent. The effect of seed priming was more prominent on seeds of both cultivars primed with CaCl2 and germinated with100 mM NaCl. It was in order of 22.4% and 16% for Hordeum vulgare and Hordeum maritimum, respectively, compared to their controls.” Was replaced by “Among the priming agent, CaCl2 seemed to be the most efficient protector causing an increase by 26 and 79% in FG%, respectively in both species. While, without priming agents the 200 mM NaCl caused 47% and 24.5% reduction of FG% for H. vulgare and H. maritimum, respectively.”
- 1,4,5,6,7,8 What do the letters (a,ab,ah,ef,c,b and another) above the posts mean?
Our answer:
These letters belong to the statistical analysis. Our objective is to compare the effect of priming agents on both barley species exposed to salt concentrations. The effect is significant when the letters are different.
- 2,3 Statistics should be added.
Our answer:
Figure 2 was deleted as proposed by your colleague and statistics were added in Figure 3 which belongs to the MDG turns in this version to figure 2.
- 2 “P CaCl2/100mM NaCl” needs to be replaced with “CaCl2/100mM NaCl”.
Our answer:
Figure 2 was deleted as proposed Reviewer 3.
- 3 “KNO3+100mM NaCl” needs to be replaced with “KNO3+100mM NaCl”.
Our answer:
As the Referee recommended, it was replaced on the figure 3 which turns in this version to figure 2.
- How do the authors explain the difference in the effects for the two barley crops? Нow do these species differ from each other?
Our answer:
The two barley species differ from each other by their origins. In fact Hordeum maritimum is a facultative halophyte, a wild cereal However Hordeum vulgare is a cultivated glycophyte. The first species is able to tolerate high salt concentration but the second one is relatively tolerant.
- 4,6,7,8,9,10 Legends should be below the diagram names.
Our answer:
As the Referee recommended, it was done for all figures. Just the number of figures changed because I deleted some of them as proposed by the Referee 3.
- 5 What is about Hordeum maritimum?
Our answer:
We presented root number just for Hordeum vulgare because Hordeum maritimum have just a principal root it is not enough to count it and analyze this parameter. Since our objective is to compare both species we preferred to delete this parameter according to your comment.
- 6, 7 What is the reason for the 10-fold difference between the dry weight of two crops?
Our answer:
The difference between the dry weights of two crops is due the fact that Hordeum maritimum tissues are smaller than Hordeum vulgare. I am comparing two species one of them is a glycophyte cultivated (Hordeum vulgare) while the other one is an halophyte wild or spontaneous (Hordeum maritimum). So, as you suggested that the following paragraph was added (see introduction)
11)
59 line “….tomato [13] and maize [14,15] .Many…” needs to be replaced with ““….tomato [13] and maize [14,15]. Many…”.
Our answer:
As the Referee noted, tomato [13] and maize [14,15] .Many…” was replaced with ““….tomato [13] and maize [14,15]. Many…”.
64 line “…sol[17,18,19] . Similar” needs to be replaced with “…sol [17,18,19]. Similar”
Our answer:
As the Referee noted, “…sol[17,18,19] . Similar” was replaced with “…sol [17, 18, 19]. Similar”
69 line “Additionally, kaya et al. revealed..” needs to be replaced with “Additionally, Kaya et al. revealed…”.
Our answer:
As the Referee noted, “Additionally, kaya et al. revealed..” was replaced with “Additionally, Kaya et al. revealed…”.
75 line “power of k+ ion” needs to be replaced with “power of K+ ion”.
Our answer:
As the Referee noted, “power of k+ ion” was replaced with“ power of K+ ion”.
564-565 lines “primed with 5mM of CaCl2 solution for 20 hours and 564 soaked with 2% of KNO3” needs to be replaced with “primed with 5 mM of CaCl2 solution for 20 hours or 564 soaked with 2% of KNO3”.
Our answer:
As the Referee noted, “primed with 5mM of CaCl2 solution for 20 hours and 564 soaked with 2% of KNO3” was replaced with “primed with 5 mM of CaCl2 solution for 20 hours or 564 soaked with 2% of KNO3”.
- 579 line “This allows clearing:” It looks strange.
Our answer:
“This allows clearing:” It was deleted as the referee noted.
- 609-610 lines Сheck ion designation. Do this throughout the text.
Our answer:
As the Referee noted, ion designation was checked.
- 619-621 Please, design what do the EC1 and EC2 mean.
Our answer:
As the Referee noted, it was done.
- 631 and 636 lines Please, check for spaces. Do this throughout the text.
Our answer:
As the Referee noted, it was done.
Answer to Reviewer 2 comments:
We thank the Reviewer for their careful reading of the manuscript and their constructive remarks. We sincerely appreciate all the valuable comments and suggestions which helped us to improve the quality of our manuscript.
Detailed Response to Reviewer 2 comments
- I am questioned by the data according to which the exposure of seeds to sodium chloride solutions is accompanied by an increase in their water content. The authors explain this by the fact that the seeds absorb salt, which leads to a drop in the water potential in the seed cells and the absorption of water. Everything that I know in this field of science, the exposure of plant objects to salt solution is accompanied by dehydration. It seems to me that the proposed explanation of the data obtained is not sufficient.
Our answer:
Our results showed that salt stress increased water content on Hordeum vulgare seeds in both tissues (Fig5). These results are consistent with those reported by Debez et al. (2004) in Cakile maritima, M’sehli et al (2011) in Medicago ciliaris and Boukari et al. (2019) in Medicago sativa who noted that WC increased under salt stress in Gabes ecotype shoots as well as roots. This phenomenon could be due the osmotic potential of cells which was probably lowered so that the water supply was not disrupted by external salinity.
It is not clear why the authors used filter paper when incubating seeds in Petri dishes. The filter paper is not so indifferent. Many years ago prof. X Borris of the University of Greifswald has shown that there is a factor in the filter paper (called the "paper factor") that affects seed germination.
Our answer:
One of our objectives is to analyze ion content, so we used a specific paper called ashless Filter paper. This type of paper doesn’t interfere the germination process and doesn’t contain element that could be absorbed by seeds.
It is necessary to unify the designation of the concentration of solutions. The work uses three salts; the concentration of one of them is expressed in moles, and the other two salts - in percent.
Our answer:
A preliminary test was carried out, in fact, for both barley species, for each pretreatment agent several concentrations and several times were tested and then according to statistics analyses the time and concentration that gave a better stimulation of the germinal parameters and the growth traits of seeds were chosen for each agent. Results of the preliminary test showed that the effect of KNO3 was more pronounced with 2% at 40h; this of KCl was more prominent with 2% at 40h, however CaCl2 was more efficient with 5mM at 20h.
Answer to Reviewer 3 comments:
We thank the Reviewer for their careful reading of the manuscript and their constructive remarks. We sincerely appreciate all the valuable comments and suggestions which helped us to improve the quality of our manuscript.
Detailed Response to Reviewer 3 comments
I think, the salt stress responses of both H. vulgare (with different accessions) and H marinum (H. maritimum) have been widely investigated even during germination, although the authors did not mention these results in the introduction. H. maritimum, a wild seaside barley, is a halophyte plant while H. vulgare is glycophyte and their salt tolerance mechanism are completely different. This information should be mentioned in the introduction.
Our answer:
As referee noted, the requested informations was added in the introduction.
-It seems that the effect of priming agents were efficient in H. vulgare (DW), while were unaffected in H. maritimum probably due to the different salt tolerance mechanisms. It should discuss better these differences.
Our answer:
As referee noted: This information was explained in the discussion part (See line )
L85-86 and L557: H vulgare and H maritimum are two species, not two cultivars. It should be given cultivar name of H. vulgare used in the experiments (as see in https://www.publish.csiro.au/fp/FP18046) . In addition, and as it was mentioned in the MM section, the seeds of H maritimum was collected from …. Are you sure that these seeds originated from genetically homogen population?
Our answer:
Our objective is to investigate the effect of priming with different agents on the germination of seeds of both barley species under salinity stress. We are focused only on the physiological responses so we collected seeds from the same territories. We are not working in genetic traits.
Although the authors presented a large number of data, these data are originated from different combinations of a germination test. Thus, it is evident that all data lead to the same conclusion. Otherwise, I do not think, that all data are necessary to present. If you think, that each figure provides new information, please discuss them in details, otherwise you can eliminate some of them (e.g. Fig.2, Fig.6,).
Our answer:
AS referee proposed, Fig 2 and Fig 6 were eliminated.
Results: The results section contains many unnecessary (long sentences with little information). Such as between L92-102; L103-107, 108, 124-126, 131, 157-161…195-207, 210-214 (What a surprise! – it is well known information), 332-336, 442-445 and so on. These should be eliminated or shortened. This type of repetitions makes the text boring and the readers lose their interest since they do not get real new information. Less is often more!
For example: L92-102: Results showed that salinity affected negatively the FG % of unprimed seeds on both barley cultivars. This effect depended on the salt concentration and the cultivar Indeed, side impacts were more pronounced when seeds were germinated with 200 mM NaCl…..:
You can write: Priming with KCl, KNO3 or CaCl2 did not modify the germination rate without salt stress, but alleviated the effect of salt stress caused by either 100 or 200 mM NaCl both in H maritimum and H. vulgare. Among the priming agent, CaCl2 seemed the most efficient protector causing only 22.4 and 16% reduction in FG%, while without priming agents the 200mM NaCl caused 47% and 24.5% reduction of FG% for H. vulgare and H. maritimum, respectively.
Instead of L157-176, you can write:
The coleoptiles and radicles length of unprimed seeds in Hordeum vulgare decreased by 65.4% and 66.7% when the seeds were exposed to 100 or 200mM salt stress, while those of Hordeum maritimum were reduced by 27% and 30.2%, respectively, compared to their controls (Fig. 4). The priming agents, especiallly CaCl2 enhanced growth of radicle in H. maritime and coleoptyle length in both species with or without salt treatement. …..
Our answer:
Results of all parameters were shortened as recommended by referee. See Results section.
If you write “This effect depended on the salt concentration and the cultivar” or “ this beneficial effect was agent and salt concentration dependent” you should prove them statistically for example by factorial analysis of variance (ANOVA)
Our answer:
This sentence of dependence was eliminated.
The statistics of Fig. 1 seems strange. On Fig.1A where is g , if there is “h” . On Fig.1B , there is no “a” , only ab. Why it is separted from “b”. Where is “A”? Etc. Please check the statistics again. I think, it is not correct to make the statistics together for both species, but present the values on separate figures (Fig.1A and B). Alternatively, write it in the Figure legends to be evident.
Our answer:
Since the objective of our study is to compare the response of primed and unprimed seeds to salinity, the statistical analysis was done together. In one hand, we bring them together to facilitate the comparison. In the other hand, to more clarify, we were obliged to divide the results in both figure a and b.
Fig.2 presents the germination % vs time (days) not LT and T50 as indicated in L104-106. In addition, why are not presented the values of KCl, KNO3 and CaCl2/0 mM NaCl? on Fig.2 and Fig.3 too.
Our answer:
As recommended by referee, values of KCl, KNO3 and CaCl2/0 mM NaCl are now presented in fig.3. Fig.2 was eliminated as the referee recommended.
Fig.5: Please use “Radicle length (cm)” and Coleoptile length (cm) in the adequate axis (A and C) of Fig.5.
Our answer:
As referee noted, it was done.
Fig.8 the data of WC seems strange. In general the WC decrease when the plants are exposed to NaCl especially in glycophyte plants. In 4.4 it is not mentioned how many plants were used for determination of WC. Perhaps the amount of plant materials was too low. I suggest repeating this type of measurement. It is quick. Alternatively, explain better how it is possible that the WC increase after NaCl in H. vulgare, but not in H. maritimum (which is probably a salt tolerant genotype). 481-485 seem seem to contradict the former lines (474-476 –it is normal phenomenon under salt stress).
Our answer:
We checked again our results, they are okay. In fact H. Vulgare is the best cultivated cereal able to tolerate salt and drought. These results are consistent with those reported by Debez et al. (2004) in Cakile maritima, M’sehli et al (2011) in Medicago ciliaris and Boukari et al. (2019) in Medicago sativa who noted that WC increased under salt stress in Gabes ecotype shoots as well as roots. This phenomenon could be due the osmotic potential of cells which was probably lowered so that the water supply was not disrupted by external salinity.
In addition, please label the Y axis more precisely (e.g. WC of radicle) . It is true for Fig.9 too.
Our answer:
Done, as referee noted.
Discussion
More explanation were added
MM section:
Please add the name of cultivar of H. vulgare used in the experiments.
Our answer:
Done
Please explain better why these concentrations (5mM CaCl2 and 2% KCl and 2% KNO3) and time (20 h or 40h) were used as priming. Why these conditions were used?
Our answer:
A preliminary test was carried out, in fact, for both barley species, for each pretreatment agent several concentrations and several times were tested and then according to statistics analyses the time and concentration that gave a better stimulation of the germinal parameters and the growth traits of seeds were chosen for each agent. Results of the preliminary test showed that the effect of KNO3 was more pronounced with 2% at 40h; this of KCl was more prominent with 2% at 40h, however CaCl2 was more efficient with 5mM at 20h. Moreover, many previous studies testing priming with those agents, most of them worked with concentration in % for KCl and KNO3and in mM for CaCl2.
4.5. I am not sure that 20mg DW (app. 200mg FW) was enough for correct determination of mineral elements.
We used 20 mg of DW and the volume of liquid of extraction is less so we were able to extract the maximum of ions. The same protocol is always used in many investigations.
4.6. Membrane integrity was estimated by measuring the MSI using a digital conductivity meter (type Metrohom). Initially, we cut 0.2 g of fresh material and placed it in a falcon with distilled water (10 ml), after then, we heated it at 32 °C for 30 minutes and finally, we recorded the electrical conductivity (EC1). The conductivity was also assayed after placing the samples at 100 °C for 2 hours (EC2). The MSI was measured according to Dionioese and Tobita using the following formula [29]: Please add the number of repetition for each measure parameters within the same subsection (not in statistics -4.8). It is difficult to follow. In addition, please add how many Petri dishes were used for each treatment.
Our answer:
Done
I think, only a germination test is insufficient for publication in Plants. I suggest following this type of experiment with older plants to demonstrate that the priming could be efficient at later developmental stages too and study what kind of mechanisms operate during the priming in H. vulgare or H. maritimum. (Although I believe that you can find information about it in many papers.
Our answer:
Our objective is to study seed physiological and biochemical response at germinative and early seedling growth and especially to evaluate the possible remedial effect of priming with different agents under salinity conditions. As you suggested, this work will be continued by an investigation at the vegetative stage.

Reviewer 3 Report
The paper studied the effect of KCl, KNO3 and CaCl2 in the germination potential of an accession of Hordeum vulgare and H. maritimum under salt stress conditions (0, 100 and 2000mM).
Improvement of germination and early seed growth by priming could be efficient only if the salt stress is temporary, but under permanent salt stress only an efficient salt tolerance mechanism (based on exclusion of NaCl from cytosol) can be appropriate. Perhaps the additive amount of KCl, KNO3 or CaCl2 resulted in higher amount of K+, Ca2+, as substrates, for transporters operating in Na+ uptake, provided amelioration of germination. However, if the NaCl is also available at later developmental stages, the KCl, KNO3 and CaCl2 are unable to protect the plants from salt stress. If plants will die later or not produce seeds, the priming become inefficient (pointless)
I think with a more accurate search you can find a lot of articles on the effect of salinity at the germination state even under priming. Look at older (published before “internet age”) papers too.
I think, the salt stress responses of both the H. vulgare (with different accessions) and H marinum (H. maritimum) have been widely investigated even during germination, although the authors did not mention these results in the introduction. H. maritimum, a wild seaside barley, is a halophyte plant while H. vulgare is glycophyte and their salt tolerance mechanism are completely different. These information should be mentioned in the introduction.
It seems that the effect of priming agents were efficient in H. vulgare (DW), while were unaffected in H. maritimum probably due to the different salt tolerance mechanisms. It should discuss better these differences.
L85-86 and L557: H vulgare and H maritimum are two species, not two cultivars. It should be given cultivar name of H. vulgare used in the experiments (as see in https://www.publish.csiro.au/fp/FP18046) . In addition, and as it was mentioned in the MM section, the seeds of H maritimum was collected from …. Are you sure that these seeds originated from genetically homogen population?
Although the authors presented a large number of data, these data are originated from different combinations of a germination test. Thus, it is evident that all data lead to the same conclusion. Otherwise, I do not think, that all data are necessary to present. If you think, that each figure provides new information, please discuss them in details, otherwise you can eliminate some of them (e.g. Fig.2, Fig.6,).
Results: The results section contains many unnecessary (long sentences with little information). Such as between L92-102; L103-107, 108, 124-126, 131, 157-161…195-207, 210-214 (What a surprise! – it is well known information), 332-336, 442-445 and so on. These should be eliminated or shortened. This type of repetitions makes the text boring and the readers lose their interest since they do not get real new information. Less is often more!
For example: L92-102: Results showed that salinity affected negatively the FG % of unprimed seeds on both barley cultivars. This effect depended on the salt concentration and the cultivar Indeed, side impacts were more pronounced when seeds were germinated with 200 mM NaCl…..:
You can write: Priming with KCl, KNO3 or CaCl2 did not modify the germination rate without salt stress, but alleviated the effect of salt stress caused by either 100 or 200 mM NaCl both in H maritimum and H. vulgare. Among the priming agent, CaCl2 seemed the most efficient protector causing only 22.4 and 16% reduction in FG%, while without priming agents the 200mM NaCl caused 47% and 24.5% reduction of FG% for H. vulgare and H. maritimum, respectively.
Instead of L157-176, you can write:
The coleoptiles and radicles length of unprimed seeds in Hordeum vulgare decreased by 65.4% and 66.7% when the seeds were exposed to 100 or 200mM salt stress, while those of Hordeum maritimum were reduced by 27% and 30.2%, respectively, compared to their controls (Fig. 4). The priming agents, especiallly CaCl2 enhanced growth of radicle in H. maritime and coleoptyle length in both species with or without salt treatement. …..
If you write “This effect depended on the salt concentration and the cultivar” or “ this beneficial effect was agent and salt concentration dependent” you should prove them statistically for example by factorial analysis of variance (ANOVA)
The statistics of Fig. 1 seems strange. On Fig.1A where is g , if there is “h” . On Fig.1B , there is no “a” , only ab. Why it is separted from “b”. Where is “A”? Etc. Please check the statistics again. I think, it is not correct to make the statistics together for both species, but present the values on separate figures (Fig.1A and B). Alternatively, write it in the Figure legends to be evident.
Fig.2 presents the germination % vs time (days) not LT and T50 as indicated in L104-106. In addition, why are not presented the values of KCl, KNO3 and CaCl2/0 mM NaCl? on Fig.2 and Fig.3 too.
Fig.4. I suggest comparing the effect of priming agents at 0, 100mM and 200mM NaCl. It is possible that the enhancements caused by priming agents are similar at 0, 100 or 200 mM NaCl, indicating that its effect is independent of salinity.
Fig.5: Please use “Radicle length (cm)” and Coleoptile length (cm) in the adequate axis (A and C) of Fig.5.
Fig.8 the data of WC seems strange. In general the WC decrease when the plants are exposed to NaCl especially in glycophyte plants. In 4.4 it is not mentioned how many plants were used for determination of WC. Perhaps the amount of plant materials was too low. I suggest repeating this type of measurement. It is quick. Alternatively, explain better how it is possible that the WC increase after NaCl in H. vulgare, but not in H. maritimum (which is probably a salt tolerant genotype). 481-485 seem seem to contradict the former lines (474-476 –it is normal phenomenon under salt stress)
In addition, please label the Y axis more precisely (e.g. WC of radicle) . It is true for Fig.9 too.
Discussion. In the discussion section, try to explain better your results:
It is still not clear why do the CaCl2 give better protection against salt stress than KCl or KNO3? .
How do the priming agents affect the Fe2+ , Mg2+ content?
What was the reason for genotypic differences?
Why the priming agents seemed more efficient at 100mM NaCl tha et 200mM in H. vulgare.
Some references related to the topic are missing:
https://www.publish.csiro.au/fp/FP18046
https://pubmed.ncbi.nlm.nih.gov/31292634/
https://pubmed.ncbi.nlm.nih.gov/31292634/
https://www.ncbi.nlm.nih.gov/pmc/articles/PMC7914899/
In addition, many typing mistakes are found in the text (e.g. k+ instead of K+), , etc.
MM section:
Please add the name of cultivar of H. vulgare used in the experiments.
Please explain better why these concentrations (5mM CaCl2 and 2% KCl and 2% KNO3) and time (20 h or 40h) were used as priming. Why these conditions were used?
4.5. I am not sure that 20mg DW (app. 200mg FW) was enough for correct determination of mineral elements.
4.6. Membrane integrity was estimated by measuring the MSI using a digital conductivity meter (type Metrohom). Initially, we cut 0.2 g of fresh material and placed it in a falcon with distilled water (10 ml), after then, we heated it at 32 °C for 30 minutes and finally, we recorded the electrical conductivity (EC1). The conductivity was also assayed after placing the samples at 100 °C for 2 hours (EC2). The MSI was measured according to Dionioese and Tobita using the following formula [29]:
Please add the number of repetition for each measure parameters within the same subsection (not in statistics -4.8). It is difficult to follow. In addition, please add how many Petri dishes were used for each treatment.
I think, only a germination test is insufficient for publication in Plants. I suggest following this type of experiment with older plants to demonstrate that the priming could be efficient at later developmental stages too and study what kind of mechanisms operate during the priming in H. vulgare or H. maritimum. (Although I believe that you can find information about it in many papers.
The results section should be significantly shortening and the discussion section should be focused better to explain the results.
Author Response

(The authors gave the same response as above.)

Round 2
Reviewer 3 Report
I read the revised version of the manuscript, the comments of reviewers (they found similar mistakes) and the answers of authors.
I found that the ms should be improved more to be able to accept for publication in Plants.
Many remarks were negligated or not answered properly. (I indicated in red in the attached file).
L94-95 is not true. It depend on the genotypes. Even the text suggests that the H. vulgare can be as salt tolerant as H. maritimum, but it is not true. The text should be evident: H. vulgare is a glycophyte plant. (it is true) and different salt tolerant mechanisms are operated in these species. Generally, the salt tolerance of glycophyte plants is worse than that of halophytes. However, it is true that many barley genotypes showed better salt tolerance than genotypes in other cereals, such as inwheat, but it depends on the genotypes. For instance barley cv. Igri is more sensitive that many Australian or Egyptian wheat genotypes, etc.
Please look at the literature more carefully:
https://www.ncbi.nlm.nih.gov/pmc/articles/PMC4332615/
https://www.intechopen.com/online-first/effects-of-salinity-on-seed-germination-and-early-seedling-stage
an so on.
L98-99 is not true.
In google it is possible to find many papers, and even several are mentioned in the discussion section. So it is not true that there is no experiment in priming under salt stress. Even a study is not being done because no one has done it before (which is not true), but because a scientific question arises. So, the end of introduction should be rewritten.
https://www.researchgate.net/publication/268428018_Effect_of_seed_priming_on_growth_of_barley_Hordeum_vulgare_by_using_brackish_water_in_salt_affected_soils
https://www.ncbi.nlm.nih.gov/pmc/articles/PMC7815140/
Osmotic adjustment is one of the main mechanisms for improving salt tolerance of plants. Accumulation of different kind of osmolytes including proline, GB and sugars in plant tissues are participated in this mechanism. They role is to reserve water status in plants by absorbing water. I suppose that priming agents can improve the water status of plants by absorbing more water) in control plants, but without rpiming and with salt stress the water status of plant is usually reduced , as was observed in H. martimum (which a is a HALOPHYTE PLANT), but increased in H. vulgare (which is a glycophyte plant). Even more, since the electrolyte leakage increased in H. vulgare indicating a membrane injury, it is question how the plants can keep water under elevated membrane damage.
I think the results of WC is not logic and perhaps something was wrong during the determination. I suggest eliminating this figure, because it can question the authenticity of the results and discussion L548-565).
Even the discussion contains many speculations which are not proved.
I can’t see significant improvemnt of the ms.